# Quantifying decision-making in dynamic, continuously evolving environments

**Maria Ruesseler[1†], Lilian Aline Weber[1,2†], Tom Rhys Marshall[2,3], Jill O'Reilly[2], Laurence Tudor Hunt[1,2]***

[1]Wellcome Centre for Integrative Neuroimaging, Department of Psychiatry, University of Oxford, Oxford Centre for Human Brain Activity (OHBA) University Department of Psychiatry Warneford Hospital, Oxford, United Kingdom; [2]Department of Experimental Psychology, University of Oxford, Anna Watts Building, Radcliffe Observatory Quarter, Oxford, United Kingdom; [3]Centre for Human Brain Health, University of Birmingham, Birmingham, United Kingdom

**Abstract** During perceptual decision-making tasks, centroparietal electroencephalographic (EEG) potentials report an evidence accumulation-to-bound process that is time locked to trial onset. However, decisions in real-world environments are rarely confined to discrete trials; they instead unfold continuously, with accumulation of time-varying evidence being recency-weighted towards its immediate past. The neural mechanisms supporting recency-weighted continuous decision-making remain unclear. Here, we use a novel continuous task design to study how the centroparietal positivity (CPP) adapts to different environments that place different constraints on evidence accumulation. We show that adaptations in evidence weighting to these different environments are reflected in changes in the CPP. The CPP becomes more sensitive to fluctuations in sensory evidence when large shifts in evidence are less frequent, and the potential is primarily sensitive to fluctuations in decision-relevant (not decision-irrelevant) sensory input. A complementary triphasic component over occipito-parietal cortex encodes the sum of recently accumulated sensory evidence, and its magnitude covaries with parameters describing how different individuals integrate sensory evidence over time. A computational model based on leaky evidence accumulation suggests that these findings can be accounted for by a shift in decision threshold between different environments, which is also reflected in the magnitude of pre-decision EEG activity. Our findings reveal how adaptations in EEG responses reflect flexibility in evidence accumulation to the statistics of dynamic sensory environments.

**\*For correspondence:**
laurence.hunt@psy.ox.ac.uk

[†]These authors contributed equally to this work

## Editor's evaluation

This important study by Ruesseler, Weber and colleagues employs psychophysical kernels and EEG reverse correlation methods to identify the decision process adjustments used to account for variations in target frequency and duration in a task in which targets emerge periodically within a continuous stimulus stream. The paper provides solid evidence for the role of leak and threshold adjustments. The paper will be of interest to researchers studying mathematical models and neurophysiological correlates of decision making and more broadly authors with an interest in the application of reverse correlation techniques for neural signal analysis.

## Introduction

Unlike in most experiments, the choices that we make in daily life rarely occur in discrete trials. Naturalistic decisions instead arise organically and continuously in dynamic environments that evolve over

time (*Huk et al., 2018*; *Hunt et al., 2021*), leading to uncertainty regarding the onset of decision-relevant changes in the environment (*Orsolic et al., 2021*; *Shinn et al., 2022*). When making decisions in such dynamic environments, more recently presented evidence should therefore usually be given greater weight and more historical evidence gradually discounted – a strategy known as 'leaky' evidence accumulation. Optimising leaky evidence accumulation involves adapting one's behaviour to the overall statistics of the environment. This can be achieved by changing the rate at which previous evidence is leaked (the 'decay') and the amount of cumulative evidence required before a categorical decision is made (the 'decision threshold') (*Glaze et al., 2015*; *Kilpatrick et al., 2019*; *Veliz-Cuba et al., 2016*).

While it is known that humans (*Ganupuru et al., 2019*; *Glaze et al., 2018*; *Harun et al., 2020*; *Ossmy et al., 2013*) and other animals (*Levi et al., 2018*; *Piet et al., 2018*) can adapt the decay and decision threshold of sensory evidence accumulation to different dynamic environments, the neural mechanisms that underlie this adaptation remain unclear. In conventional trial-based paradigms, neurophysiological correlates of perceptual decision-making have been well characterised, particularly signals that resemble an evidence accumulation-to-bound process (*Gold and Shadlen, 2007*; *Hanks and Summerfield, 2017*; *O'Connell and Kelly, 2021*). Two of the best-studied human electroencephalographic (EEG) correlates of decision formation are an effector-independent centro-parietal positivity (CPP) that shows accumulator-like dynamics during decision formation (*Kelly and O'Connell, 2013*; *O'Connell et al., 2012*; *Pisauro et al., 2017*; *Twomey et al., 2015*), and motor preparation signals that emerge prior to a response (*Donner et al., 2009*; *Steinemann et al., 2018*; *Wyart et al., 2012*). Both signals can adapt their properties according to the overall statistics of a task. For example, CPP amplitude at the time of making a response is increased by emphasising speed over accuracy (*Steinemann et al., 2018*), while pre-trial motor lateralisation can reflect the prior expectation of a leftward or rightward action in the upcoming trial (*de Lange et al., 2013*; *Kelly et al., 2021*). Yet it remains unclear whether and how these signals reflect the ability to behaviourally adapt the decay of past information in continuous (dynamic) environments, which are more akin to many decisions faced in naturalistic settings.

Perhaps one reason why decision-making in dynamic environments has been less studied than trial-based choice is the uncertainty concerning how best to analyse the time-varying neural data. For example, without clearly defined discrete trials, it appears unclear to which timepoint data should be epoched. As the stimulus is continuously changing, it is also ambiguous how to disentangle responses to previous versus current sensory evidence, which may be overlapping in time. However, recent innovations in trial-based task design (*Brunton et al., 2013*; *Cheadle et al., 2014*; *Orsolic et al., 2021*) and unmixing of overlapping EEG responses (*Crosse et al., 2016*; *Ehinger and Dimigen, 2019*; *Hassall et al., 2021*; *Smith and Kutas, 2015*) have suggested potential solutions to some of these challenges. By tightly controlling how sensory evidence fluctuates over time, it becomes possible to relate moment-to-moment stimulus fluctuations to subsequent behavioural and neural responses. In addition, by using data analysis techniques that explicitly target overlapping neural responses, it is also possible to establish the temporal response function (TRF) to each new fluctuation in a continuous sensory evidence stream (*Gonçalves et al., 2014*). By combining these two approaches, we hypothesised that we would be able to characterise decision-related EEG responses in a continuous and dynamic setting, even in the absence of repeated experimental trials.

In this study, we examine how the EEG response to evidence fluctuations during a continuous perceptual decision task is affected by the overall statistics of the sensory environment. Participants were trained to attend to a continuously changing sensory evidence stream, in which brief 'response periods' were embedded that were reported via buttonpress. We demonstrate a CPP-like potential that is sensitive to each fluctuation in the stream of continuous sensory evidence, a motor preparation signal prior to buttonpress, and a triphasic occipito-parietal component that reflects the integrated sum of recently presented evidence. We then show that subjects' behaviour adapts appropriately to different sensory environments, and that changes in centroparietal and motor preparation prior to buttonpress signals reflect adaptation of leaky sensory evidence integration to different environments. This is not a simple feature of adaptation to sensory surprise as the CPP-like potential largely responds to decision-relevant, not decision-irrelevant, evidence fluctuations.

We also show substantial between-subject variability in the decay time constant of the 'integration kernel', a measure that reflects the structure of evidence that is presented prior to a participant's

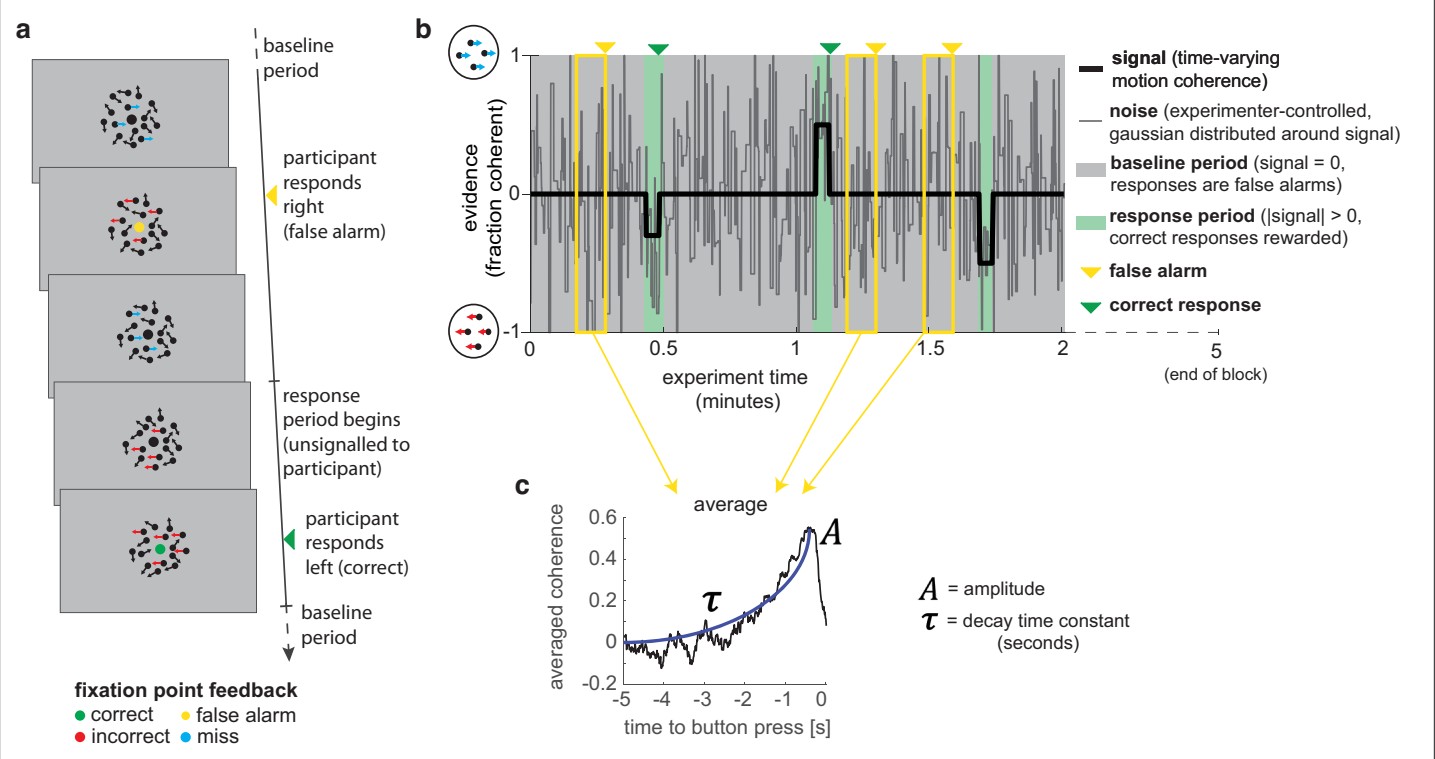

**Figure 1.** A novel, continuous version of the random dot kinematogram (RDK) paradigm allows empirical measurement of participants' leaky evidence integration kernels in dynamic environments. (**a**) Task design. Participants continuously attend to a centrally presented RDK stimulus, for 5 min at a time. They aim to successfully report motion direction during 'response periods' (when coherent motion signal is on average non-zero) and withhold responding during 'baseline periods' (when signal is on average zero). (**b**) Task structure (example block; response periods are 'rare'). During both baseline (grey) and response periods (green), the signal (black line) is corrupted with experimenter-controlled noise (grey line). The noise fluctuations that precede each response (arrows) can be averaged to obtain the evidence integration kernel. (**c**) The resulting evidence integration kernel for false alarms is well described by an exponential decay function, whose decay time constant in seconds is controlled by the free parameter τ. The equation for this kernel is in the main text, and details of kernel fitting are provided in 'Methods'.

response. This behavioural measure correlates across subjects with a neural measure of evidence accumulation: it predicts the amplitude of the triphasic centroparietal TRF to absolute recent sensory evidence. We show via computational modelling that these changes in integration kernels are most likely explained via a change in the decision threshold of a leaky evidence accumulator. Collectively, these results provide a neural characterisation of human decision-making in a dynamic, continuously evolving perceptual environment and how this can adapt to the overall statistics of the environment.

## Results

### A novel task for exploring behavioural and EEG adaptations to the statistics of dynamic sensory environments

To study evidence accumulation in a continuous setting, we designed a novel variant of the classic random dot kinematogram (RDK) paradigm (*Britten et al., 1992*; *Donner et al., 2009*; *Kelly and O'Connell, 2013*; *Newsome and Paré, 1988*). Subjects continuously monitored a stream of time-varying sensory evidence (hereafter referred to as 'motion coherence') for blocks of 5 min (*Figure 1a*). During extended 'baseline periods' (grey shaded area in *Figure 1b*), the average level of motion coherence (black line) in the stimulus was zero, whereas during shorter intermittent 'response periods' (green shaded area), the mean level of motion coherence became non-zero (either 30, 40, or 50% motion coherence). The participants' task was to report whenever they detected such a period of coherent motion using a left or right buttonpress. Importantly, the onset of response periods was not explicitly signalled to the participant. If they responded accurately (during a response period or within 500 ms of it ending), they received a reward (+3 points); if they failed to report a response

period ('missed response period'), or they responded during a baseline period ('false alarm'), they received a small punishment (–1.5 points). Participants also received a larger punishment (–3 points) if they reported the incorrect motion direction during a response period; in practice, such errors were very rare. Feedback was presented by changing the colour of the central fixation point for 500 ms (*Figure 1a*), and they were trained on the meaning of these colours as part of extensive pre-experiment training (see 'Methods'). The accumulated total points were then converted into a monetary pay-out at the end of the task. Participants completed six runs, each consisting of four 5 min blocks; they were given a short break between each block and a longer break between runs.

Crucially, the net motion presented to the participant on each frame of the stimulus was not the *average* level of motion coherence (black line in *Figure 1b*), but instead was a noisy sample from a Gaussian distribution about this mean (grey line). This noisy sample was resampled on average every 280 ms (inter-sample interval drawn from an exponential distribution, truncated at 1000 ms). This 'experimenter-controlled sensory noise' confers several benefits.

First, the injection of sensory noise places a stronger demand on temporal evidence integration than a classical RDK task. This is because any individual period of strong motion coherence could be driven by a noisy sample during a baseline period, rather than necessarily signalling the onset of a response period (*Figure 1b*). As such, continuous and temporally extended integration is essential to successfully disambiguate changes in the mean from noisy samples around the baseline. Indeed, participants would occasionally make 'false alarms' during baseline periods in which the structure of the preceding noise stream mistakenly convinced them they were in a response period (see Figure 3, below). Indeed, this means that a 'false alarm' in our paradigm has a slightly different meaning than in most psychophysics experiments; rather than it referring to participants responding when a stimulus was *not present*, we use the term to refer to participants responding when there was no shift in the *mean signal* from baseline.

Second, the noise fluctuations allow a 'reverse correlation' approach to studying subjects' evidence integration. Simply by averaging the noisy stimulus that was presented prior to each response, we could extract an 'integration kernel' that empirically reveals how far back in time the motion coherence is being integrated – in other words, how quickly previous motion is decaying in the participant's mind – and how strong this motion coherence needed to be on average to support a choice. We performed this reverse correlation for both false alarm responses (example shown in *Figure 1c*; these responses are well described by an exponential decay function detailed below) and correct responses. The fact that integration kernels naturally arise from false alarms, in the same manner as from correct responses, demonstrates that false alarms were not due to motor noise or other spurious causes. Instead, false alarms were driven by participants treating noise fluctuations during baseline periods as sensory evidence to be integrated across time, and the motion coherence that preceded 'false alarms' need not even distinguish targets from non-targets.

Finally, and perhaps most importantly, the experimenter-controlled sensory noise allows us to characterise how continuous sensory evidence fluctuations cause changes in the simultaneously recorded continuous EEG signal. To study this, we used a deconvolutional general linear model (GLM) approach (*Crosse et al., 2016*; *Ehinger and Dimigen, 2019*; *Gonçalves et al., 2014*; *Hassall et al., 2021*) to estimate TRFs to various events relating to the time-varying sensory evidence. We describe this approach and the resulting TRFs in more detail below.

## Behavioural adaptations to environments with different statistical properties

We used this paradigm to investigate whether and how participants adapted their evidence integration behaviour to the overall statistics of the sensory environment. To test this, we manipulated both the duration and frequency of 'response periods' in the task. We hypothesised that this would affect the decay of past sensory evidence and/or the decision threshold used to commit to a response. Importantly for our subsequent analyses, we kept the generative statistics of the Gaussian noise during 'baseline periods' consistent across conditions. This allowed us to directly compare behavioural evidence integration kernels for false alarms and EEG TRFs across conditions without any potential confound from how the noise was structured.

Within each 20 min run, participants completed four pseudorandomly ordered 5 min blocks drawn from a 2 * 2 factorial design (*Figure 2a*). Response periods were either LONG (5 s) or SHORT (3 s),

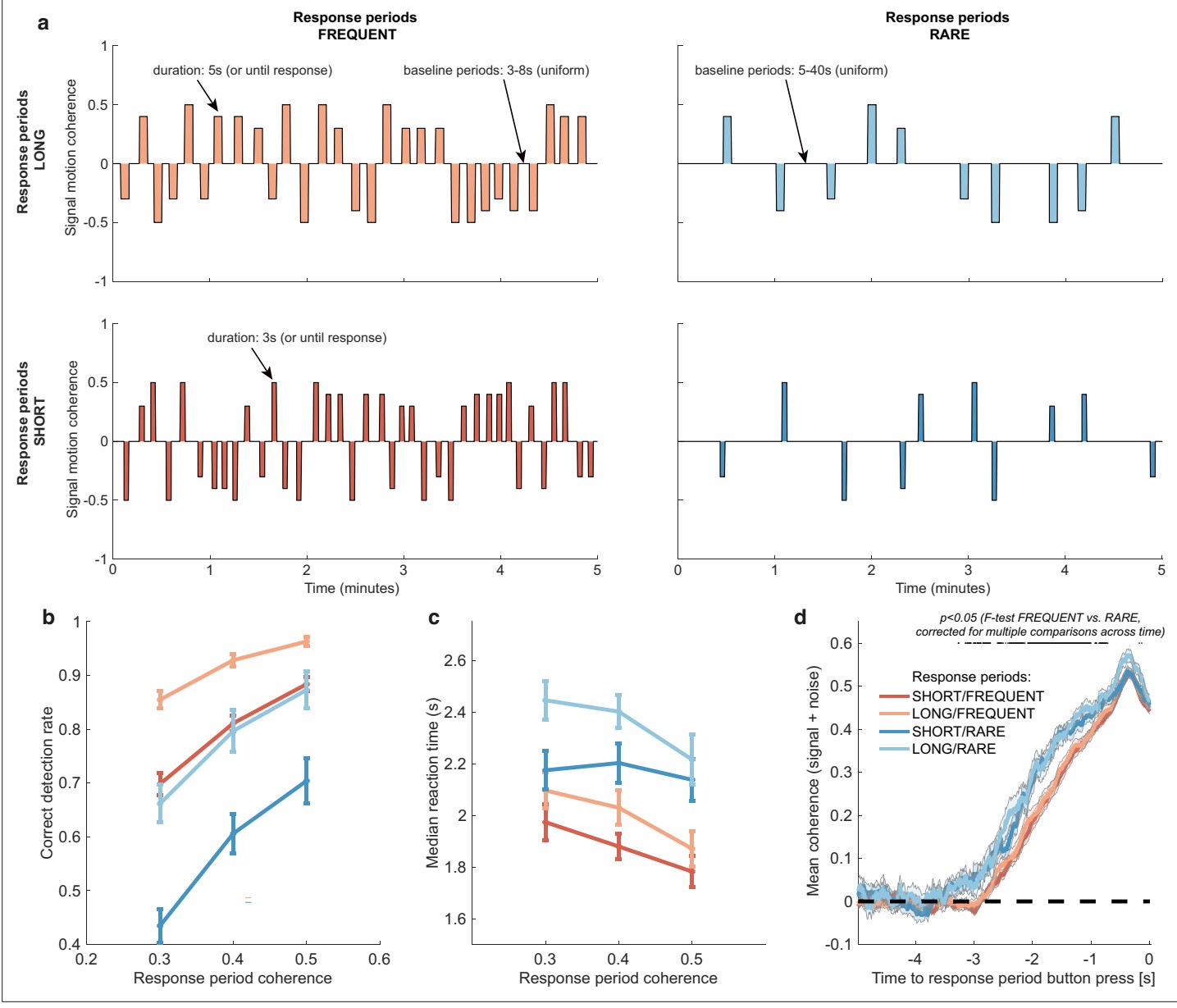

**Figure 2.** Variations in response period structure across different environments elicit behavioural adaptations in decision-making. (**a**) Structure of response periods (signal only, before noise was added to the stimulus stream) across the different environments. This was manipulated in a 2 * 2 design, where response periods were either FREQUENT or RARE, and LONG or SHORT. Participants were extensively trained on these statistics prior to the task, and the current environment was explicitly cued to the participant. (**b**) Correct detection rate for all response periods. Participants successfully detected more response periods when they were LONG than SHORT (as would be expected, because the response period is longer), but also detected more when they were FREQUENT than RARE. (**c**) Median reaction time (time taken to respond after start of response period) for successfully reported response periods across the four conditions. Participants took longer in RARE versus FREQUENT conditions, and in LONG versus SHORT conditions. (**d**) Integration kernels for 'response periods' shows a main effect of FREQUENT versus RARE response periods, but unexpectedly no effect of LONG versus SHORT response periods. See main text for further discussion of this analysis. All plots in (**b–d**) show mean ± s.e. across 24 participants. Note that to make reaction times and integration kernels comparable between the four conditions, we only include those responses that were shorter than 3.5 s in analyses for (**c**) and (**d**) (i.e. the maximum response time in SHORT response periods).

The online version of this article includes the following figure supplement(s) for figure 2:

**Figure supplement 1.** Logistic mixed model of subjects choices during response periods, with regressors of mean motion coherence (avgCoh), variance of motion coherence (cohVar), response period Frequency (trlFrq), response period length (trlLen), and interaction terms between these regressors.

and either FREQUENT (baseline periods between 3 and 8 s in duration) or RARE (baseline periods between 5 and 40 s). Participants were extensively trained on these trial statistics prior to completing the task and were then explicitly cued which environment they were currently in. As a consequence, participants neither had to learn nor infer the higher-order statistics of the sensory environment during the task; instead, they had to adapt their decision behaviour according to the pre-learnt statistics of the cued environment.

## Response periods

We first tested whether participants adjusted their behaviour across the four conditions by analysing detection behaviour for response periods. We used a three-way repeated-measures ANOVA to test for effects of motion coherence, response period length, and response period frequency.

As expected, participants were faster ($F_{(2,46)} = 29.51$, $p=7.45 * 10^{-9}$) and more accurate ($F_{(2,46)} = 17.6$, $p=0.00035$) at detecting response periods with a higher level of average motion coherence (*Figure 2b and c*).

Examining the effects of response period frequency, participants were less likely ($F_{(1,23)} = 41.99$, $p=1.30 * 10^{-6}$) and slower ($F_{(1,23)} = 83.24$, $p=6.23 * 10^{-9}$) to detect response periods when these were RARE (blue lines in *Figure 2b and c*) than when they were FREQUENT (red lines in *Figure 2b and c*). We hypothesised that this could be explained by participants requiring a stronger overall level of recently accumulated evidence before committing to a response when response periods were rare. Indeed, when we examined the 'integration kernel' of average sensory evidence for successful responses, significantly more cumulative evidence was required for RARE than FREQUENT response periods, stretching back up to 3.5 s prior to the commitment to a response (*Figure 2d*; *F*-test with permutation-based correction for multiple comparisons, $p<0.05$).

We also found that participants detected response periods more frequently when these were LONG than when they were SHORT (*Figure 2b*; $F_{(1,23)} = 178.52$, $p=2.51 * 10^{-12}$). This is unsurprising as there was simply more time to detect the change in motion during these response periods. In fact, participants were slightly more conservative in their responding when trials were LONG, as shown by a longer average reaction time for LONG response periods relative to SHORT (even when restricting this analysis to focus on LONG responses that were less than 3.5 s, the maximum possible SHORT response duration including 500 ms response tolerance period; *Figure 2c*; $F_{(1,23)} = 70.00$, $p=2.78 * 10^{-8}$). This was also reflected in their false alarm frequency, as shown below. Surprisingly, the sensory evidence integration properties (i.e. the 'integration kernels', calculated by averaging the signal prior to the decision, collapsing across all levels of mean motion coherence) were *not* affected by the length of response periods (*Figure 2d*). This ran contrary to our initial hypothesis that participants would integrate evidence for longer when response periods were LONG. We suggest that this may result from the manipulation of response period duration being relatively small (3 s versus 5 s) compared to the manipulation of response period frequency. We also note that the significant difference between FREQUENT and RARE trials in *Figure 2d* should not be over-interpreted as it could be influenced by RT differences (*Figure 2c*) and the associated shift in the onset of the signal contribution and/or the difference in average coherence detection across conditions (*Figure 2b*). Importantly, we control for these confounds below by examining the integration kernels to false alarms (in the absence of changes in mean signal).

We also considered an alternative stimulus detection strategy of changes in stimulus variance across time rather than changes in stimulus mean. This hypothesis relied upon the fact that response periods had smaller standard deviations in the Gaussian noise distribution than baseline periods – a stimulus feature that we introduced to avoid excessive samples of 'maximal' (100%) motion coherence when the mean was non-zero. To test whether the variance of the stimulus might also affect participants' detection, we performed a logistic mixed effects model on participants' responses (*Figure 2—figure supplement 1*). Detection probability was the dependent variable, and mean motion coherence, variance of motion coherence, response period frequency, and length were independent variables, along with interaction terms. We found that stimulus variance during response periods did indeed impact detection probability; response periods with a higher variance in motion coherence were less likely to be detected. Crucially, however, the main effects of mean motion coherence, trial frequency, and trial length (equivalent to the effects plotted in main *Figure 2b*) were left unaffected by the inclusion of this coregressor.

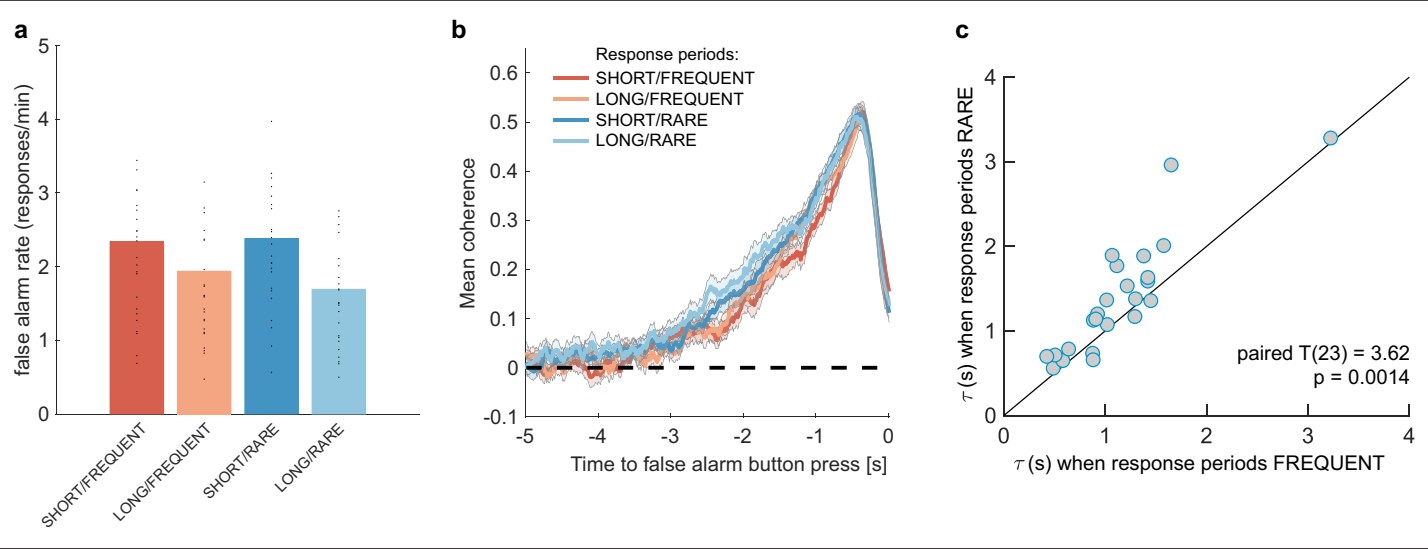

**Figure 3.** Changes in false alarm response frequency and evidence integration kernels across environments with different statistical structure. (**a**) False alarm rates (responses during baseline periods) showed a main effect of response period duration – participants showed significantly lower false alarm rates when response periods were LONG versus SHORT (F(1,23) = 58.67, p=8.98 * 10⁻⁸). This is consistent with having a more cautious response threshold (also evidenced by longer reaction times during response periods, see *Figure 2c*), although it could also be interpreted as shorter response periods inducing more confusion between signal and noise. (**b**) Integration kernels calculated for false alarms across the four conditions. Lines show mean +/- s.e. across 24 participants. (**c**) Exponential decay model fitted to individual participants' kernels during false alarms shows a significantly longer decay time constant when response periods were RARE versus FREQUENT. The data points show the time constant, $\tau$, for each participant after fitting a model of exponential decay to the integration kernel. The equation for this kernel is in the main text, and details of kernel fitting are provided in 'Methods'.

The online version of this article includes the following figure supplement(s) for figure 3:

**Figure supplement 1.** Between-subject variability in evidence integration kernels exceeds between-condition variability.

## False alarms

We then examined whether false alarms differed across conditions, testing for effects of response period frequency and length on false alarm rate using a two-way repeated-measures ANOVA. We found that despite the structure of the noise stream being identical across the four conditions, there was a lower overall frequency of false alarms in LONG versus SHORT conditions (*Figure 3a*; F(1,23) = 58.67, p=8.98 * 10⁻⁸). This provides further evidence that participants were overall more conservative in their responses in LONG conditions than SHORT. (In other words, for an equivalent level of sensory evidence, the participants were less likely to make a response.) There was no effect of response period frequency on false alarm rate (F(1,23) = 0.37, p=0.55).

We then examined what *caused* participants to false alarm during baseline periods. Were participants still integrating evidence continuously during these periods of the task, or might false alarms be driven by other spurious factors, such as motor noise? We tested this by calculating integration kernels derived from these responses. We found the recovered evidence integration kernels showed exponential decay weighting, implying that participants were indeed performing continuous evidence integration throughout baseline periods as well as response periods, and that evidence accumulation was more temporally extended when response periods were RARE rather than FREQUENT (*Figure 3b*). The slight differences in integration kernels between *Figure 3b* and *Figure 2d* (shorter duration, and return to baseline close to the response) are due to the inclusion of the average motion signal in *Figure 3b*, rather than just the noise.

## Between-participant variation in evidence integration

We then sought to characterise the time constant of leaky evidence integration within each individual participant. To do this, we fit an exponential decay model (*Figure 1c*) to the empirical integration kernel from false alarm responses:

$$k(t) = Ae^{\frac{-t}{\tau}}$$

where $k(t)$ is the height of the integration kernel $t$ seconds before its peak; $A$ is the peak amplitude of the integration kernel (in units that denote the fraction of dots moving towards the chosen response direction); and $\tau$ is the decay time constant (in units of seconds). We note that this exponential decay model is theoretically motivated by the leaky evidence accumulation model, which implies that past evidence will leak from the accumulator with an exponential decay (*Bogacz et al., 2006*).

Our exponential decay model provided a good fit to data at a single-subject level (median $R^2 = 0.82$, 95% confidence intervals for $R^2 = [0.42, 0.93]$; see *Figure 3—figure supplement 1* for example fits), as demonstrated by the strong reliability across conditions for both $A$ and $\tau$ (*Figure 3c*, *Figure 3—figure supplement 1a*; Pearson's correlation between SHORT and LONG conditions: $\tau$: R(23) = 0.71; A: R(23) = 0.72; Pearson's correlation between RARE and FREQUENT conditions: $\tau$: R(23) = 0.87; A: R(23) = 0.81; all p<0.0001). Indeed, a striking feature of these integration kernels was that variation across *individuals* exceeded the variation observed across *conditions* (e.g. see *Figure 3c*).

Consistent with our earlier analyses (*Figures 2d and 3b*), we found that by fitting this single-subject model, $\tau$ was significantly longer when response periods were RARE than FREQUENT (paired T(23) = 3.62, p=0.0014; *Figure 3c*) but A did not differ between these conditions (paired T(23) = 0.03, p=0.97; *Figure 3—figure supplement 1a*). Again consistent with our analyses of behaviour during response periods (*Figure 2d*), there was no difference between these parameters for LONG versus SHORT response periods ($\tau$: paired T(23) = 0.82, p=0.42; A: paired T(23) = -0.97, p=0.34; *Figure 3—figure supplement 1a*).

In summary, these results indicate that participants adapted to response periods being rarer by accumulating sensory evidence with a longer time constant of integration, but that there was also substantial between-subject variability in evidence accumulation across participants.

## Computational modelling of leaky evidence accumulation

We next considered what adjustments within a computational model of leaky evidence accumulation might account for the behavioural adaptation across different environments, and the variability across participants. We simulated a well-established model of leaky evidence accumulation, the Ornstein–Uhlenbeck process (*Bogacz et al., 2006*; *Brunton et al., 2013*; *Ossmy et al., 2013*). Here, evidence is accumulated over time according to

$$X_t = (1 + \lambda)X_{t-1} + gM_t + \varepsilon_t$$

where is a parameter that (when constrained to be negative) determines the *leak* of past sensory evidence in the decision variable; $g$ is a parameter that determines the *gain* applied to the momentary sensory evidence at each timepoint $M_t$ ; and $_t$ denotes Gaussian-distributed white noise with mean 0 and variance σ. The model emits a response every time that a decision *threshold* $\pm\theta$ is exceeded, at which point $X_{t+1}$ is reset to 0. Note that if were set to 0 rather than negative, this model would be equivalent to the widely used Drift Diffusion Model, in which previously accumulated evidence is perfectly retained in the decision variable $X_t$ . Such a model would be inappropriate in the current paradigm as the structure of the task demands that past sensory evidence should gradually be discounted.

We first considered what adjustments of the model parameters governing leak and decision threshold, and $\theta$, would lead to optimal performance in terms of points gained across the entire block (*Figure 4*). We simulated model behaviour using a range of possible values of these parameters, while holding $g$ constant and assuming σ is primarily a property of low-level sensory processing and so also remains constant. The optimal parameterisation of the Ornstein–Uhlenbeck process depended upon the number of correct responses/missed trials in response periods versus the number of false alarms during baseline periods (*Figure 4—figure supplement 1*). For example, setting the response threshold $\theta$ to a low value (e.g. <1 in *Figure 4—figure supplement 1*) leads to the model correctly detecting virtually all response periods, but also emitting so many false alarms that the total points obtained would be negative. By contrast, setting the threshold slightly higher still allows for correct responses, but they now outnumber false alarms, meaning that the model accumulates points across the block.

*Figure 4a* shows the area of model performance that performs in the top 10% of all parameterisations that we considered, and *Figure 4b* shows the best parameterisations for 30 streams of evidence

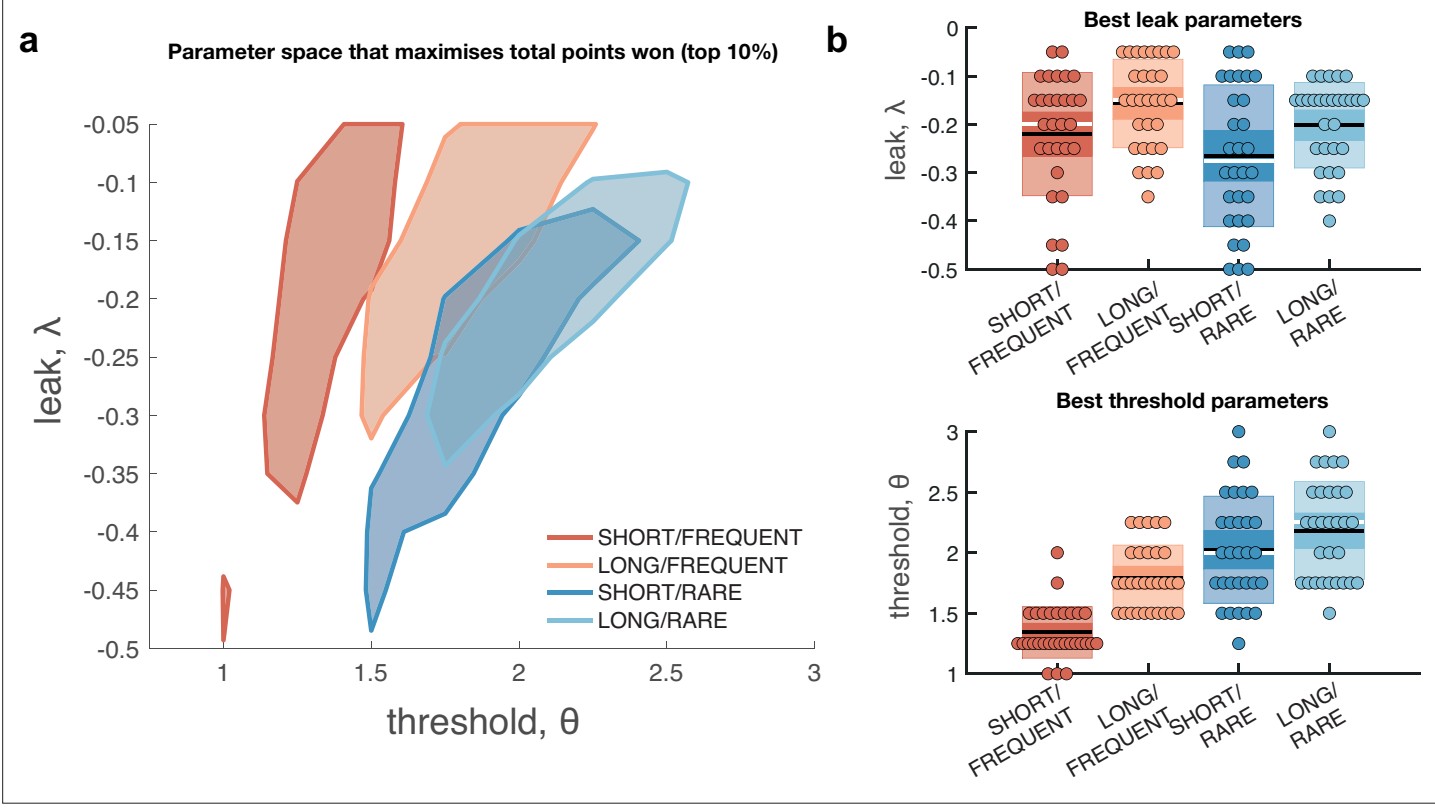

**Figure 4.** Optimal leak and threshold for a leaky accumulator model differs as a function of task condition. (**a**) We performed a grid search over the parameters and *θ* to evaluate the performance (points won) for different parameterisations (*Figure 4—figure supplement 1*). The shaded area denotes the areas of model performance that lay in the top 10% of all models considered. The optimal area differs across conditions, and the optimal setting for leak and threshold co-vary with one another. (**b**) We used the evidence stream presented to each participant (each dot = one 5 min block), to identify the model parameterisation that would maximise total reward gained for each subject in each condition (see also *Figure 4—figure supplement 2*).

The online version of this article includes the following figure supplement(s) for figure 4:

**Figure supplement 1.** Grid search across parameter space for Ornstein–Uhlenbeck process.

**Figure supplement 2.** The optimal model parameters for leak and threshold correlate with each other.

that were presented to participants in the task. Notably, the optimal decision threshold *θ* was *traded off* against the optimal setting for leak (*Figure 4a*, *Figure 4—figure supplement 2*). In other words, a model in which past sensory evidence leaked more rapidly (i.e. was more negative) could be compensated by a decrease in *θ*, to retain a high level of overall points gained. This led to a 'ridge' in parameter space where a given set of values for and *θ* would provide high task performance. The location of this ridge differed across the four environments, implying that participants would indeed need to adapt these parameters across conditions.

We confirmed the optimal settings for and θ by presenting the actual stimulus streams that were presented to our participants, and identifying the values of these two parameters that maximised total points won (*Figure 4b*). This demonstrated that when response periods were LONG rather than SHORT, the optimal adjustment was to reduce the amount of leak in the model, so that incoming sensory evidence persisted for longer within the decision variable. When response periods were RARE rather than FREQUENT, the model could be optimised by increasing the decision threshold. This, in turn, would make the model more conservative, consistent with the reduced detection rates and accuracy shown in *Figure 2b and c*. (We note, however, that this is slightly inconsistent with the pattern of behavioural false alarm rates shown in *Figure 3a*. We suggest that this may be under the control of further factors such as time-varying urgency [*Geuzebroek et al., 2022*], something we do not consider in the current model.)

If changes in θ are primarily driven by the frequency of response periods, and changes in are primarily driven by their length, then how can we explain the fact that the decay time constant *τ* of

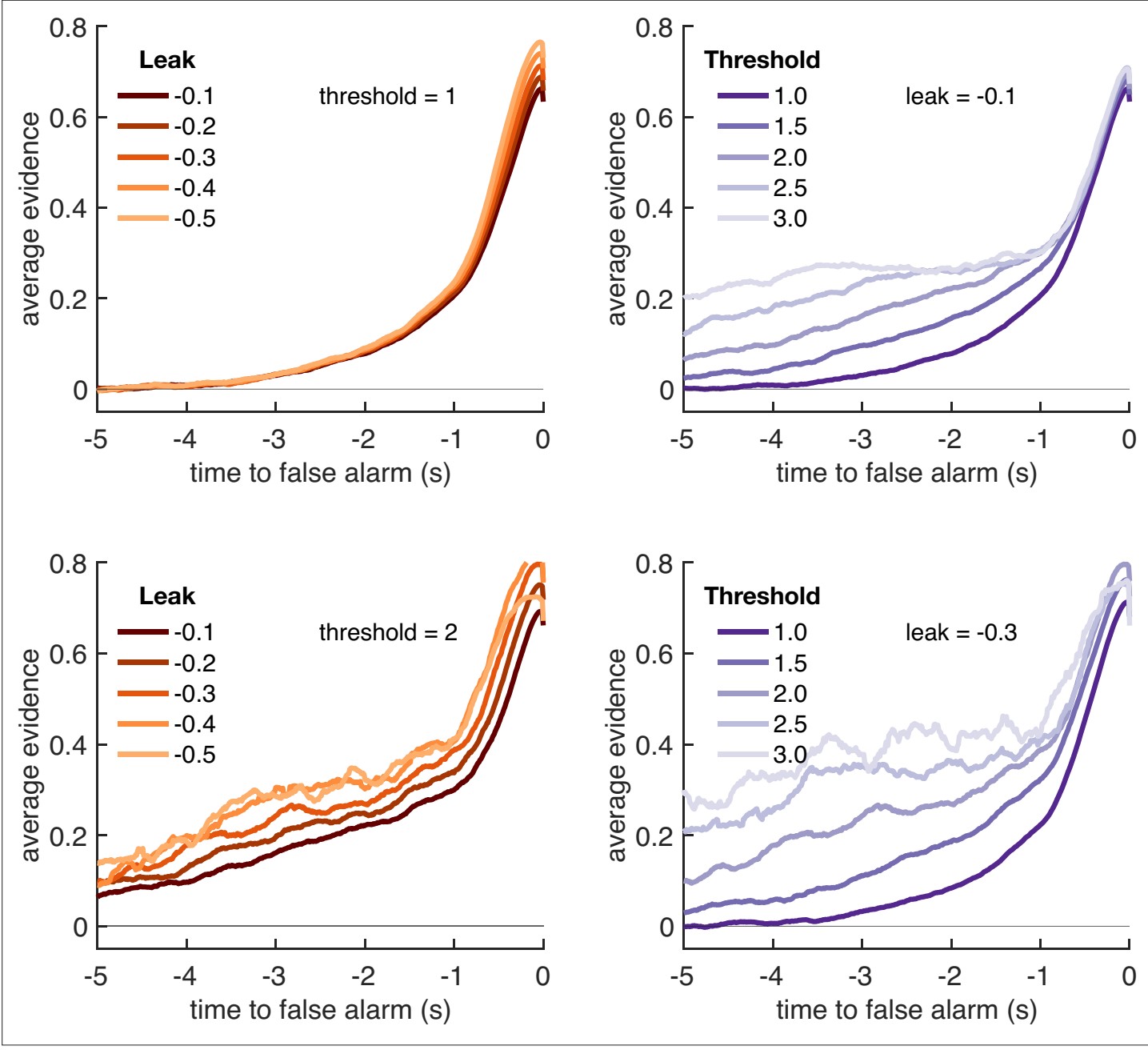

**Figure 5.** Variation in model threshold primarily determines decay time constant of integration kernels. We performed an integration kernel analysis on false alarms emitted by the Ornstein–Uhlenbeck process with different settings for leak ($\lambda$, left column) and threshold ($\theta$, right column) while holding the other parameter constant. Variation in $\theta$ would invariably affect the requirement for temporally sustained evidence to emit a false alarm (a higher threshold requiring sustained evidence); variation in primarily affect the amplitude of the eventual kernel.

integration kernels is affected by frequency but not length? To answer this, we performed an equivalent analysis of integration kernels on our model simulations. We epoched and averaged the sensory evidence that preceded each response made by the decision model and examined the effects of and $\theta$ on the recovered integration kernels (**Figure 5**). Surprisingly, we found that $\theta$, not $\lambda$, was primarily responsible for the recovered decay time constant $\tau$. At first sight, this appears counterintuitive because is directly responsible for the decay of past sensory evidence in the model of leaky accumulation. However, this is counteracted by the fact that the only data that enters this analysis is when the model has passed decision threshold, and a response is emitted – if the threshold is set higher, then a consistent stream of positive evidence is required before threshold will be reached, producing the

effect shown in *Figure 5*. By contrast, we found that the amplitude of the integration kernel *A* was affected by manipulations of both $\theta$ and $\lambda$.

In summary, our conclusions from the computational modelling are threefold: (i) our manipulations of response period frequency and length elicited different settings for model threshold and leak respectively to maximise reward (*Figure 4b*); (ii) the 'ridge' in parameter space that performed well (top 10%) for each condition showed a trade-off between threshold and leak (*Figure 4a*, *Figure 4— figure supplement 1*), and may explain how different participants could show very different integration kernels (*Figure 3—figure supplement 1*) while still performing well on the task; and (iii) the between-condition and between-subject variation in integration kernel time constants $\tau$ is principally driven by variation in response threshold, $\theta$ (*Figure 5*).

## EEG correlates of continuous sensory evidence integration

Having established behavioural differences in evidence integration across individuals and across environments with different statistical structures, we then examined how participants' EEG responses reflected these differences. To test this, we examined the effects of the *noise fluctuations* on the EEG signal during baseline periods. We focussed on this time period for three reasons: (i) the generative

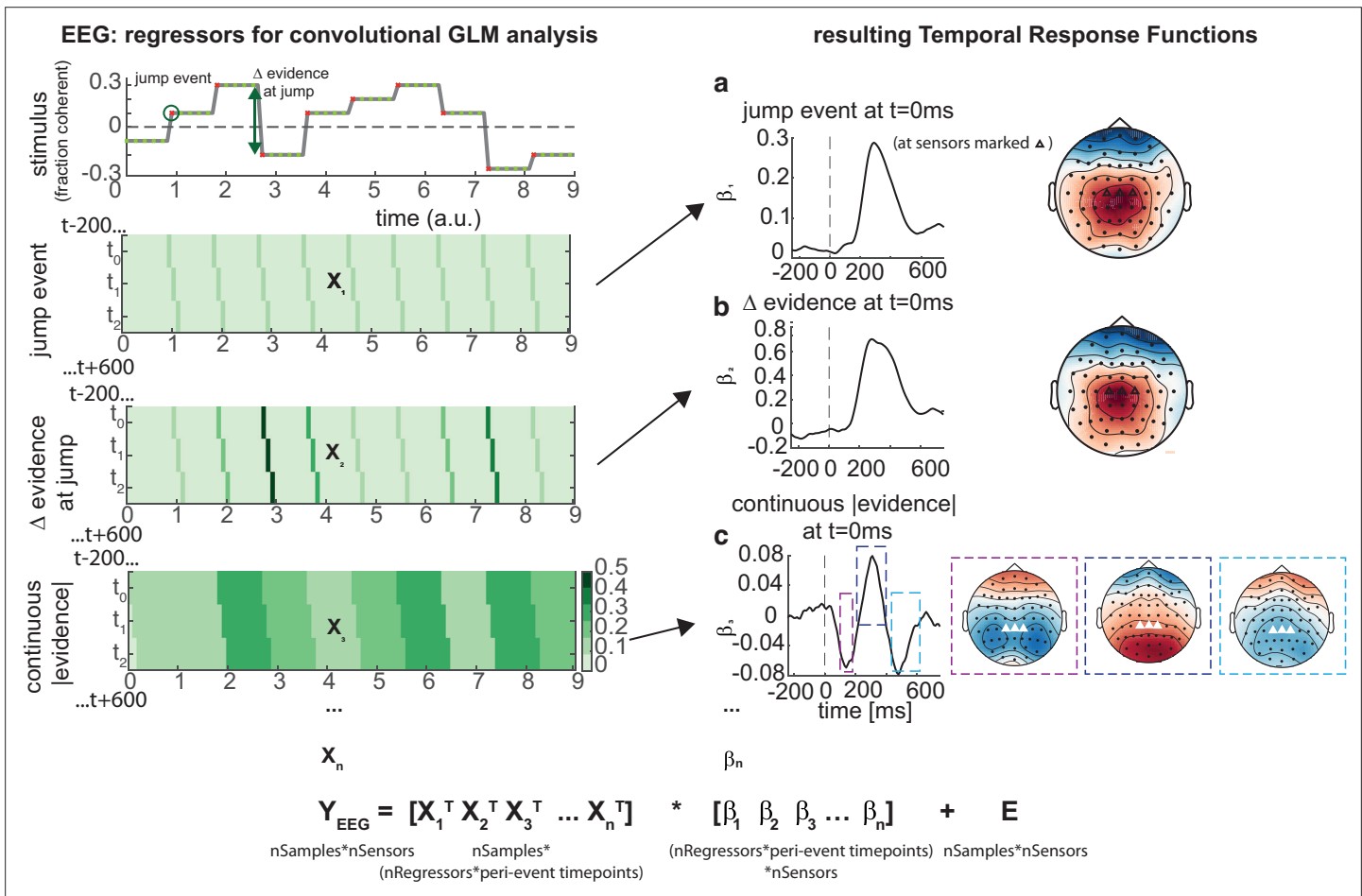

**Figure 6.** Deconvolutional general linear model to estimate electroencephalographic (EEG) temporal response functions to continuous, time-varying decision regressors. The left-hand side of the figure shows an example evidence stream during the baseline period (note that inter-sample intervals are shown as fixed duration for clarity, rather than Poisson distributed as in the real experiment). Three example regressors are shown: (**a**) 'jump event', when there was a change in the noise coherence level; (**b**) '|Δ evidence|', reflecting the magnitude of the jump update at each jump event; and (**c**) continuous |evidence|, reflecting the continuous absolute motion strength. For each of these regressors, a lagged version of the regressor timeseries is created to estimate the temporal response function (TRF) at each peri-event timepoint. This is then included in a large design matrix *X*, which is regressed onto continuous data *Y* at each sensor. This leads to a set of temporal response functions for each regressor at each sensor, shown on the right-hand side of the figure. The timecourse for each regressor shows the average regression weights at the three sensors highlighted with triangles on the scalp topography. Full details of the design matrix used in our analysis of the EEG data are provided in 'Methods'.

statistics of the noise were identically matched across all four task conditions; (ii) behavioural evidence from 'false alarms' clearly indicated that participants were still integrating sensory evidence during baseline (*Figure 3*, *Figure 3—figure supplement 1*); and (iii) the large number of noise fluctuations embedded in the stimulus (>1000 per 5 min block) meant that we had many events of interest to recover EEG TRFs with a high signal-to-noise ratio (*Gonçalves et al., 2014*; *Lalor et al., 2006*).

We therefore built a deconvolutional GLM to estimate TRFs to various events relating to the time-varying noise fluctuations during baseline periods. In particular, this GLM included regressors that described (*Figure 6*) (a) 'jump events' in the experimenter-controlled noise ('stick functions' that were 1 whenever the motion coherence changed, and 0 elsewhere); (b) the 'change in evidence' associated with each jump event (stick functions with a parametric modulator of |Δevidence|, i.e. absolute difference between previous and current motion coherence); and (c) the current |evidence| (a continuous regressor, reflecting the absolute difference from 0 across time; note that this regressor is absoluted to look for effector-independent signals processing current motion strength, as opposed to those signed towards leftward/rightward motion). We also included several further regressors to capture EEG correlates of the onset of response periods, the level of motion coherence, and correct and false alarm buttonpresses (see 'Methods' for full details).

Using this approach, we found a set of consistent TRFs that reliably reflected the continuous updates in the time-evolving sensory evidence during baseline (*Figure 6*). In particular, the two regressors that reflected changes in the sensory evidence ('jump events') and the magnitude of |Δevidence| both elicited positive-going scalp topographies over centroparietal electrodes, peaking ~300 ms after this change occurred (*Figure 6a and b*). This scalp topography, timecourse, and reporting of |Δevidence| are consistent with the P300 component (*Donchin, 1981*; *Duncan-Johnson and Donchin, 1977*; *Mars et al., 2008*; *Squires et al., 1976*). The scalp topography is also consistent with the CPP (*Kelly and O'Connell, 2013*; *O'Connell et al., 2012*; *O'Connell and Kelly, 2021*), whose ramp-to-threshold dynamics have been proposed to account for many established effects in the P300 literature (*Twomey et al., 2015*). In addition, the continuous |evidence| regressor elicited a triphasic potential over centroparietal electrodes (*Figure 6c*). This triphasic potential is notably similar to EEG potentials reflecting 'decision update' signals during trial-based tasks that require integration of multiple, discrete pieces of evidence (*Wyart et al., 2012*).

## Increased CPP responses to Δevidence and response thresholds when response periods are rare

We then examined whether these TRFs to noise fluctuations were adapting across the different sensory environments. Given our behavioural findings concerning integration kernels (*Figures 2d and 3b*), we reasoned that we would most likely identify differences as a function of response period frequency rather than length. Indeed, we found that the centroparietal response to the same change in sensory evidence was larger when response periods were RARE than when they were FREQUENT (*Figure 7a*, *Figure 7—figure supplement 2*; p=0.017, cluster-based permutation test). As large changes in mean evidence are less frequent in the RARE condition, the increased neural response to |Δevidence| may reflect the increased statistical surprise associated with the same magnitude of change in evidence in this condition. In addition, when making a correct response, preparatory motor activity over central electrodes reached a larger decision threshold for RARE versus FREQUENT response periods (*Figure 7b*; p=0.041, cluster-based permutation test). We found similar effects in beta-band desynchronisation prior, averaged over the same electrodes; beta desynchronisation was greater in RARE than FREQUENT response periods. As discussed in the computational modelling section above, this is consistent with the changes in integration kernels between these conditions as it may reflect a change in decision threshold. It is also consistent with the lower detection rates and slower reaction times when response periods are RARE (*Figure 2b and c*), which also imply a higher response threshold. By contrast, we found no statistically significant difference for either of these regressors between SHORT versus LONG response periods (*Figure 7—figure supplements 1 and 2*). We also found qualitatively similar results for false alarm responses.

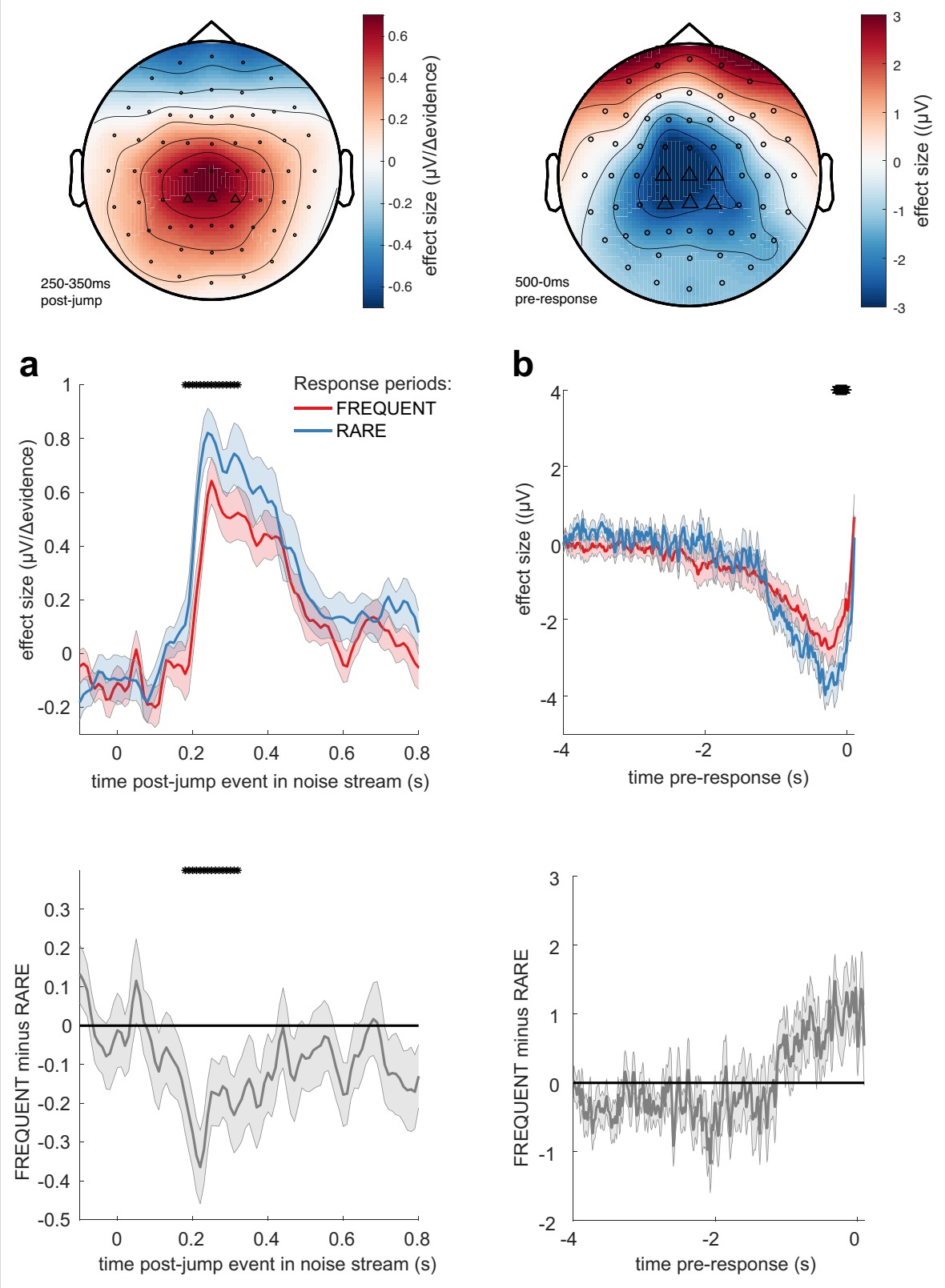

**Figure 7.** Adaptations of electroencephalographic (EEG) responses to sensory environments where response periods are RARE versus FREQUENT. (**a**) Centroparietal electrodes (see triangles in scalp topography) showed a significantly greater response to Δevidence during 'jump events' in the noise stream when response periods were RARE than when they were FREQUENT. (**b**) Central and centroparietal electrodes showed a significantly greater negative-going potential immediately prior to a buttonpress during response periods. Lines and error bars show mean ± s.e.m. across 24 participants.

*Figure 7 continued on next page*

*Figure 7 continued*

* (solid black line at top of figure) denotes significant difference between FREQUENT and RARE (p<0.05, cluster corrected for multiple comparisons across time). Details of the permutation testing used for multiple-comparisons correction are provided in 'Methods'.

The online version of this article includes the following figure supplement(s) for figure 7:

**Figure supplement 1.** No significant differences in electroencephalographic (EEG) responses between conditions where response periods were SHORT versus LONG.

**Figure supplement 2.** Individual subject electroencephalographic (EEG) effects for the Δevidence regressor over centroparietal electrodes from 200 to 400 ms after the change in evidence.

### Responses to |Δevidence| reflect decision-relevant, not decision-irrelevant, statistics of stimulus

Given the potential role of |Δevidence| in surprise detection, we next asked whether the centroparietal response to |Δevidence| reflected low-level sensory properties of changes in the motion stimulus, or higher-level signals relevant to decision-making. To test this, we collected an additional control dataset where the stimulus contained both horizontal motion (decision-relevant) that subjects had to integrate, as in the main experiment, but also vertical motion (decision-irrelevant) that had the same low-level sensory statistics. As in the main experiment, we found that centroparietal responses reflected both 'jump events' and their associated |Δevidence| for decision-relevant motion, but these were substantially reduced for regressors that reflected changes in decision-irrelevant evidence (*Figure 8*). This implies that low-level sensory surprise alone does not account for the centroparietal responses to |Δevidence| in our continuous paradigm. Instead, the neural response is better described as reporting change detection that is relevant to signal detection and discrimination. It is possible that such change detection would be useful to indicate when a response period is more likely to arise in the task (*Shinn et al., 2022*).

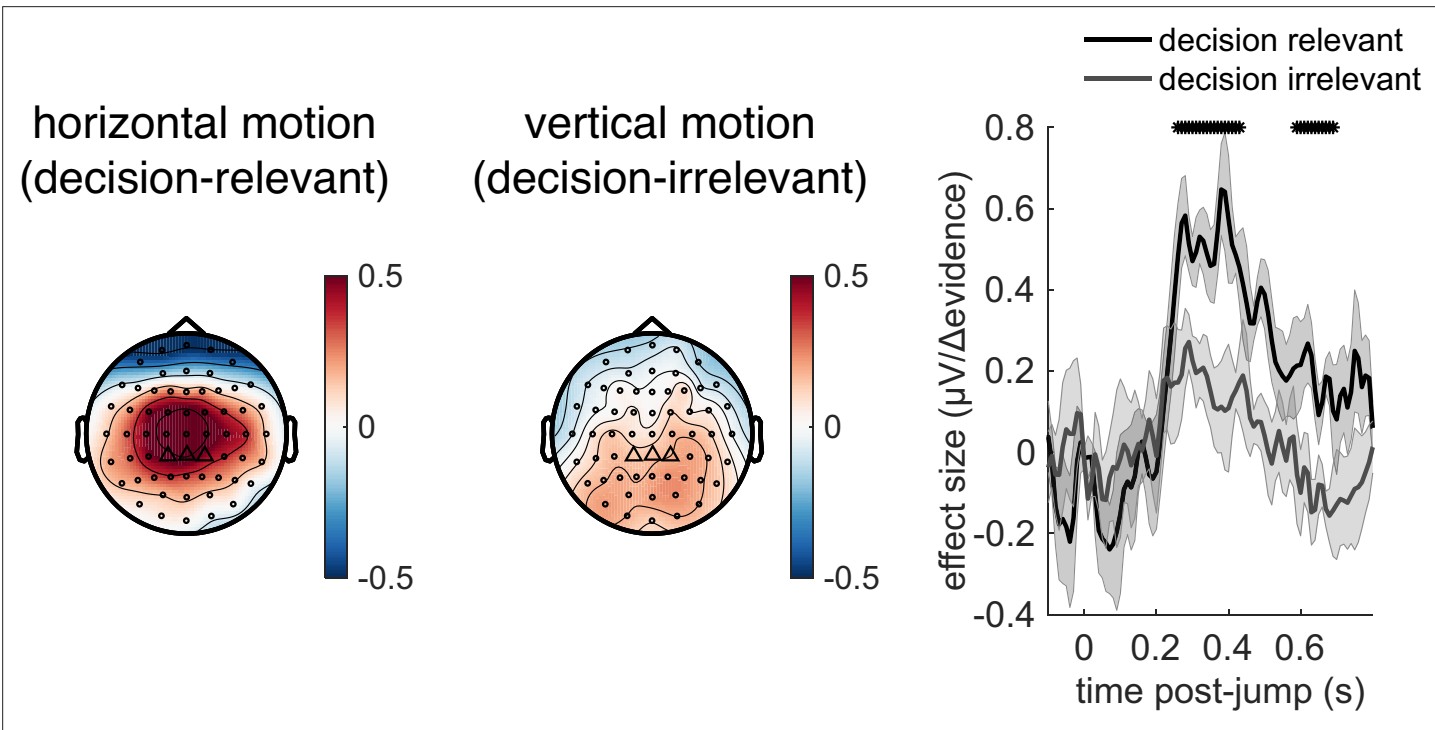

**Figure 8.** Control experiment demonstrates that response to |Δevidence| is primarily found to decision-relevant horizontal motion, but not decision-irrelevant vertical motion (with identical generative statistics). Lines show mean +/- s.e.m. across 6 participants. * denotes timepoints where the response to |Δevidence| is significantly greater for decision-relevant motion than decision-irrelevant motion, while controlling for multiple-comparisons across time (see 'Methods').

## Behavioural-neural correlation between evidence integration kernels and TRFs to continuous sensory evidence

Finally, given the consistency and between-subject variability in integration time constants shown in *Figure 3c* and *Figure 3—figure supplement 1*, we explored whether any components relating to processing of sensory evidence might reflect *cross-subject* variation in evidence integration. We therefore performed a behavioural-neural correlation between participants' integration time constants $\tau$ and their TRFs to sensory noise fluctuations. (Note that the integration time constants were fit using

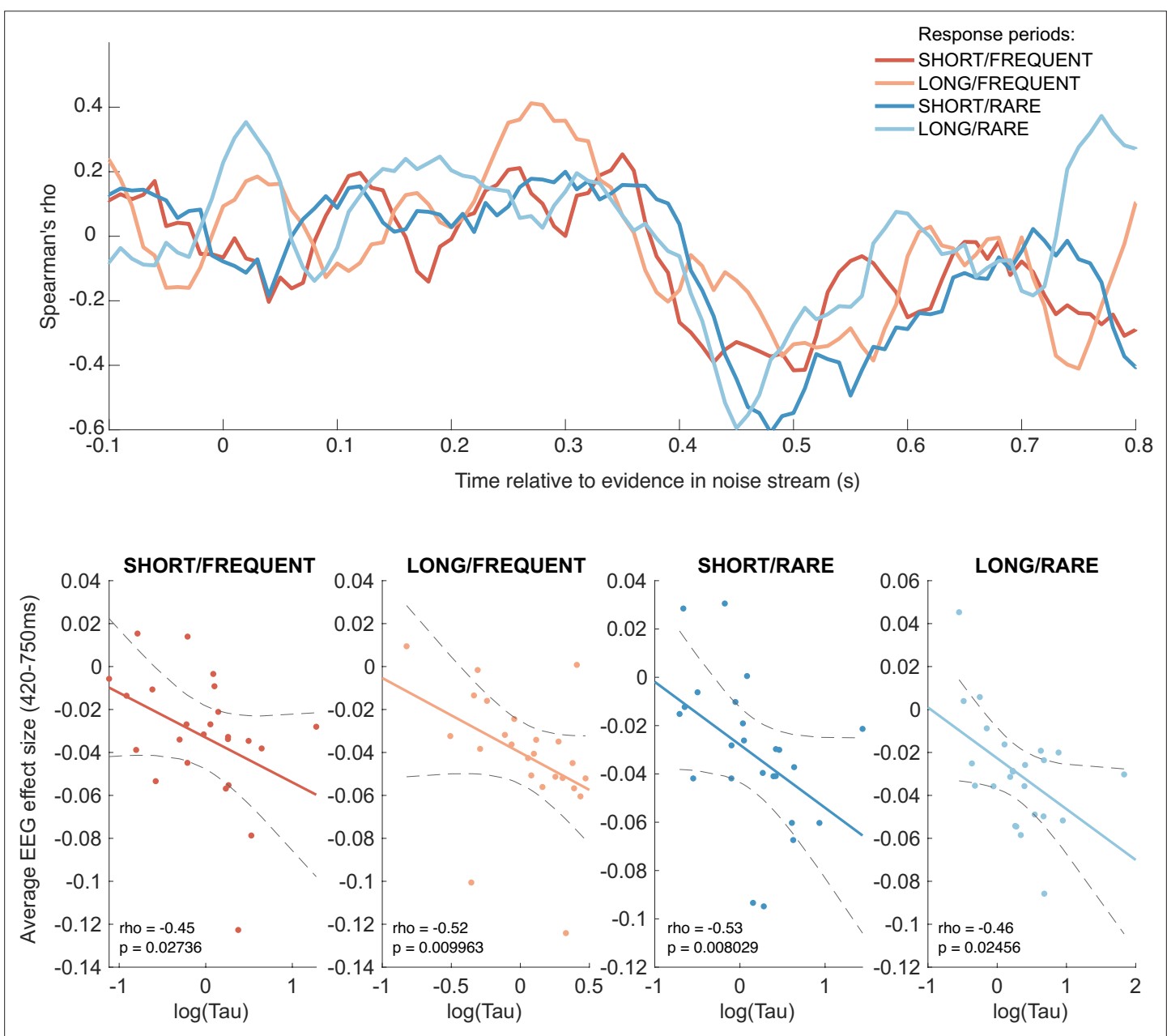

**Figure 9.** Behavioural-neural correlation (across subjects) of integration decay time constant and response to absolute sensory evidence in stimulus (see *Figure 6c*). Top panel shows Spearman's rank correlation between the time-varying electroencephalographic (EEG) beta for absolute sensory evidence and individual subjects' $\tau$ parameter, separately for each of the four conditions. The negative-going correlation found in all four conditions from ~420 ms onwards coincides with the third, negative-going limb of the triphasic response to absolute sensory evidence shown in *Figure 6c*. Bottom panels show the correlation plotted separately for each of the four conditions. We plot the average EEG effect size against $\log(\tau)$ to allow for a straight-line fit (lines show mean ± 95% confidence intervals of a first-order polynomial fit between these two variables); we used Spearman's rho to calculate the relationship, as it does not assume linearity.

the equation described above, fit [using the approach described in 'Methods'] separately to the empirical integration kernels from each of the four conditions.)

We found such a correlation for the triphasic potential elicited by the continuous 'absolute sensory evidence' regressor (see *Figure 6c*). From approximately 420 ms onwards, the amplitude of the final, negative component of this component showed a negative correlation with $\tau$ across participants (*Figure 9*). In other words, this negative-going component was *larger* in amplitude (i.e. *more* negative) in participants who would integrate sensory evidence over *longer* durations (i.e. had a *higher* value of $\tau$). We suggest that this may be consistent with variation in the encoding strength of previously studied correlates of continuous decision evidence. For example, Wyart et al. found a positive centro-parietal potential 500 ms after decision information that positively encoded the current sample, but negatively encoded adjacent samples (*Wyart et al., 2012*); our finding extends this work to explore variation in the response across participants.

Although this across-subject correlation was discovered via exploratory analyses, it replicated across all four independent conditions, substantially increasing the likelihood of it being a true positive result (response periods FREQUENT and SHORT: Spearman's $\rho$ = –0.45, p=0.027; FREQUENT and LONG: $\rho$ = –0.52, p=0.0099; RARE and SHORT: $\rho$ = –0.53, p=0.0080; RARE and LONG: $\rho$ = –0.46, p=0.025). By contrast, we found no evidence for an across-subject correlation between fitted integration decay time constants and EEG regressors encoding evidence updates at jump events (Δevidence), nor between fitted amplitude parameters (*A*) and any EEG regressors.

## Discussion

Real-world decisions typically demand continuous integration of incoming evidence in dynamic environments, without external cues as to when evidence integration should be initiated or responses made. This contrasts with a long-standing tradition in decision-making and psychophysics research to confine decisions to discrete trials, which are typically externally cued to the participant. In this article, we developed an approach to measure how participants weighed their recent history of sensory evidence when performing a continuous perceptual decision task. We found that participants' behaviour was well described by an exponentially decaying integration kernel, and that participants adapted the properties of this process to the overall statistics of the sensory environment (*Ossmy et al., 2013*) across four different experimental conditions. We also found that there was substantive inter-individual variability in leaky evidence integration, with some participants rapidly discounting past evidence and others integrating over several seconds. We then demonstrated that both sources of variability (between-condition and between-subject) were reflected by changes in centroparietal EEG responses to time-varying fluctuations in sensory evidence.

Understanding the neural mechanisms supporting simple perceptual decision-making remains a key goal for the neurosciences, and the scalp topographies and signals that we have identified in this study match well with known findings in the literature on changepoint detection and sensory evidence accumulation. In particular, the well-known P300 component has long been argued to be associated with detection of statistical surprise (*Donchin, 1981*; *Duncan-Johnson and Donchin, 1977*; *Mars et al., 2008*; *Squires et al., 1976*) or, more recently, a correlate of a continuous time-evolving decision variable (*Twomey et al., 2015*) that is equivalent to the CPP (*Kelly and O'Connell, 2013*; *O'Connell et al., 2012*). Our centroparietal responses to |Δevidence| (*Figure 6b*) are consistent with a changepoint detection account of continuous decision-making (*Booras et al., 2021*), in which decision-relevant input (*Figure 8*) is evaluated for a change in latent state from a baseline period to a response period (*Nassar et al., 2019*). This account would also explain why these signals are enhanced when response periods are rarer as a large |Δevidence| is more statistically surprising when response periods are rare than when they are common. Such changepoint detection may be useful to transiently suppress neuronal activity and attend to salient incoming sensory evidence to guide choices (*Shinn et al., 2022*). Alongside this, we also identify a triphasic potential that is similar in timecourse and scalp topography to previous studies of evidence accumulation (*Wyart et al., 2012*), sensitive to the continuous incoming sensory evidence (*Figure 6c*). Although we did not find variation in this potential across conditions, we did find that its amplitude reliably predicted the time constant of leaky evidence accumulation across participants (*Figure 9*). This suggests a key role for this component in translating incoming sensory evidence into a continuous representation of the decision variable across time, but further work is needed to understand how this variable then supports

the sensorimotor transformation into a final commitment to making a choice (*Steinemann et al., 2018*). Relatedly, it is unclear what precise functional role of the pre-response central potential and centralised beta-band signals may play. One possibility is that they may reflect a change in decision threshold between RARE and FREQUENT conditions (*Figure 7b*). Yet we were unable to detect any equivalent change in lateralised beta power (i.e. a signal related to the formation of a specific choice *Hunt et al., 2013*; *Kirschner et al., 2023*). One alternative possibility is that the central ERP is a readiness potential (*Schurger et al., 2021*).

Our work sits within a broader trend in recent research of moving away from trial-based designs towards continuous decision paradigms (*Huk et al., 2018*). In addition to being more naturalistic, a key advantage of such paradigms is that it is possible to relate time-varying properties of the continuous input to continuous output variables. In 'tracking' paradigms, for example, not only is the sensory input continuous and time-varying but also the behavioural responses made by participants. This approach has been used to dramatically reduce the length of time needed from individual participants, meaning that experiments that previously required many thousands of trials over several hours to recover psychophysical functions can now be completed in a matter of minutes (*Bonnen et al., 2015*; *Knöll et al., 2018*; *Straub and Rothkopf, 2021*). In our paradigm, the behavioural responses remained discrete and sparse (as participants were completing a signal detection/discrimination task, rather than a tracking task), but the EEG data is continuous and time-varying, and our analysis approach similarly benefits from being able to relate continuous variations in sensory input to this continuous neural signal. This has been shown in previous work to yield considerable improvements in signal-to-noise ratio compared to traditional trial-based event-related potentials (*Dimigen and Ehinger, 2021*; *Gonçalves et al., 2014*; *Lalor et al., 2006*), meaning that our approach should allow us to characterise the neural response to sensory evidence integration in less recording time than in previous work. Indeed, when we examined neural responses from individual participants in this study, we found a high degree of individual-subject reliability. This efficiency is due to the high density of events in the continuous task design (>1000 jump events in each 5 min block), limiting the amount of 'dead time' present in the experimental design (*Henson, 2007*). In sum, this leads to a more efficient experimental design than in conventional trial-based experiments and may potentially allow for more rapid and reliable estimation of single-subject responses.

An intriguing finding in this work is the substantive variability in integration decay time constants across individuals. Indeed, such inter-individual variability exceeded the between-condition variability that was observed due to our experimental manipulations. We consider two possible explanations of this inter-individual variability. The first is that it is a stable, trait-like feature of sensory evidence integration that is not unique to our task, but instead reflects true variability in perceptual evidence integration across individuals. Such a hypothesis would imply that it would predict variability in integration time constants in other domains (e.g., auditory evidence integration [*Brunton et al., 2013*; *Keung et al., 2019*; *McWalter and McDermott, 2018*] or more broadly cognitive tasks that involve continuous maintenance and manipulation of information across time in working memory). If so, it may also be possible to relate variability in behavioural time constants to underlying neurobiological causes by measuring the resting autocorrelation structure of neural activity, for example, in MEG or fMRI data (*Cavanagh et al., 2020*; *Manea et al., 2022*; *Raut et al., 2020*).

An alternative hypothesis is that the individual variability we observe may be a consequence of the prior expectations that our participants have about the overall task structure, combined with learning over the course of training. One result in support of this hypothesis comes from the modelling shown in *Figure 4*. Not only does the result in *Figure 4a* show that behaviour should be adapted across different conditions, but it also shows that different individuals might potentially achieve similar performance by ending up at very different locations in this parameter space. This could in turn explain why between-subject variability in these kernels exceeded between-condition variability (*Figure 3c*, *Figure 3—figure supplement 1*). During training, different participants could have optimised their parameters to maximise points gained, but in doing so ended up at different locations on the 'ridge' of parameters shown in *Figure 4a*. To adapt behaviour between conditions, they may have then made a small adjustment in these parameters to optimise performance for each environment.

Further work will be needed to distinguish these explanations of between-subject variability in integration kernels and test competing models of participant behaviour. Although the Ornstein–Uhlenbeck process that we use is an appropriate and widely used model of the task, alternative models

might also consider a dynamically changing threshold as a function of progress through the inter-trial interval (*Geuzebroek et al., 2022*); or consider tracking the mean and variance of the stimulus over time, rather than just the mean (*Bill et al., 2022*). In this work, we also did not directly fit parameters of the Ornstein–Uhlenbeck process to participant behaviour. Although progress has recently been made in model fitting for decision-making in continuous decision-making paradigms (*Geuzebroek et al., 2022*), a key feature of our paradigm is that many responses result from the structured noise that we inject into the sensory evidence stream, which complicates the use of aggregate measures such as reaction time quantiles for model fitting. Model estimation could potentially be improved by having continuous behavioural output, as recently demonstrated in tracking paradigms (*Huk et al., 2018*; *Straub and Rothkopf, 2021*).

Our findings of behavioural adaptations according to the overall statistics of the sensory environment are consistent with findings from previous research that have examined the same question in the absence of neural measures (*Ossmy et al., 2013*; *Piet et al., 2018*). *Ossmy et al., 2013* used a trial-based paradigm with similarities to ours, involving unpredictable signal detection in the context of time-varying background noise in combination with a manipulation of signal-period duration. They used this to show that participants adapted the time constants of evidence integration in different environments. In our paradigm, we benefitted from being able to directly recover empirical integration kernels as opposed to estimating them in a model, and we also found behavioural differences that resulted from varying response period duration. However, we surprisingly found little effect on integration kernels from this manipulation; we instead found that our manipulation of response period frequency had a greater effect on the weighting of past sensory evidence. We hypothesise that this difference between our results and those of Ossmy et al. may simply result from our manipulation of response period duration not being sufficiently large (3 s versus 5 s) to require a substantial change in evidence weighting to optimise rewards. This is also consistent with our neural results, where the between-condition variation in responses to time-varying evidence was primarily found as a function of response period frequency, rather than duration.

In conclusion, our work demonstrates that it is possible to accurately measure the timecourse and neural correlates of sensory evidence integration in continuous tasks, and how this adapts to the overall properties of the environment. This work provides a framework for future work to investigate how evidence integration is adapted to other features of decision tasks, and how this may vary across individual participants. Our approach will also be useful for future work to investigate how the properties of evidence integration change in clinical populations and how they are affected by various interventions (e.g. pharmacological, electrical/magnetic stimulation, cognitive training).

## Methods
### Task design

In the continuous task, participants observed a stream of randomly moving dots in a circular aperture (*Figure 1*). A fraction of these dots move coherently to the left or to the right; the motion coherence is the proportion of dots moving in the same direction, whereas the other dots move randomly. In this study, the coherence varied between –1 (all dots move to the left) and 1 (all dots move to the right). At 0 coherence, all dots move randomly.

Unlike in trial-based versions of the task, and even previous RDM tasks where motion is continuous (e.g. *Kelly and O'Connell, 2013*), during the present task the coherence *changed* constantly. During 'baseline' periods, the *average* of these constantly changing values remains 0. Within this stream of constantly changing coherence, there were response periods in which the *average* coherence was either to the left or to the right (see *Figures 1b and 2a*). The aim of the subject performing this task was to detect these stable periods of predominantly leftward or rightward motion (response periods). For participants, this means that they should not respond as soon as they think they know in which direction the dots are moving coherently, as is the case for a discrete trial version of the RDM task. Instead, participants must weigh the recent history of motion directions to detect periods where the average motion direction of the dots was consistently leftwards or rightwards.

Participants indicated their decision about the average motion direction by pressing keyboard button 'L' to report a response period with average rightward motion and 'A' to report a response period with average leftward motion. Every time a button was pressed, a change in colour of the

central fixation point provided feedback: correct responses were indicated by a green fixation point, a red fixation point followed an incorrect response during the response period, and false alarms (i.e. buttonpresses during baseline periods) were indicated by a yellow fixation point. Whenever a response period was missed (no response made), the fixation point turned blue after 500 ms (note that a buttonpress within these 500 ms was still counted as a correct response, to account for non-decision time and allow participants to integrate over the entire length of the response period). Following correct or incorrect responses made during a 'response period', the response period was terminated immediately, and the stimulus returned to baseline.

Participants were rewarded for correct responses but lost points for any other response. They received 3 points for correct responses, punished with –3 points for incorrect responses, and missed response periods or false alarms were both punished with –1.5 points. A reward bar was shown at the end of each 5 min block to indicate how many points participants have won in total (the reward bar was shown continuously onscreen during training, but not during task performance to avoid distraction). As participants won more points, their reward increased to the right until they hit the right border of the reward bar (equivalent to a net gain of 15 points), the bar was reset to the middle of the screen and they received £0.50 bonus to take at the end of the experiment. In rare cases where participants were performing poorly and losing points on average, they hit the left border of the reward bar (–15 points) and had £0.50 deducted from their take-home bonus.

## Structure of the noise

An essential feature of this task paradigm was the noise structure, which leads to continuously varying coherence levels. Notably, the noise was placed under experimental control rather than randomly generated, meaning that we could examine how fluctuations in the noise impact participants' behavioural and neural data. More generally, the noise can be described as a series of short intervals that vary in duration and coherence (*Figure 1b*). The interval duration was sampled from an exponential distribution with a mean duration of 270 ms. This distribution was then truncated, with a minimum duration of 10 ms and a maximum duration of 1000 ms for each step. The level of motion coherence at each step was sampled randomly from a normal distribution. The mean of this normal distribution depends on whether the step occurred during baseline or a response period. During a baseline period, the mean of the normal distribution was 0. That means it was equally likely that negative or positive coherences were drawn. During response periods, the mean of the normal distribution was sampled uniformly from the set [-0.5, –0.4, –0.3, 0.3, 0.4, 0.5]. Any samples that exceeded 100% motion were set to be [+1, –1]. To limit the number of times this occurred, we set the standard deviation of the distribution to 0.3 for response periods and 0.5 for baseline periods. (We note that this could allow a strategy of tracking changes in the variance in the stimulus as well as the mean, something that we address in the supplementary note.).

## Design of the random dot motion stimulus

The task was coded in Psychtoolbox (*Brainard, 1997*; *Kleiner et al., 2007*; *Pelli, 1997*) and parameters were chosen similar to *Shadlen and Newsome, 2001*. Participants were seated 87 cm in front of the screen. Moving dots were displayed in a circular aperture subtending with a radius of 5 visual degrees on a Dell monitor with a refresh rate of 100 Hz. Dots had a size of 0.1 visual degrees and were displayed with a density of 2.5 dots per squared visual degree. The fixation point in the centre of the screen had a size of 0.3 visual degrees. All dots were black and displayed on a mid-grey background (rgb: 0.5, 0.5, 0.5).

Dots were equally divided into three sets. These sets were shown sequentially, meaning only one set per frame was shown. Each time a set reappeared on the screen the coherence on that frame dictated, the likelihood of that dot either being displaced randomly or in the direction of the coherence. Randomly displayed dots moved like Brownian motion particles, with no particular speed. Dots that moved coherently were displaced according to a speed of 7° per second (*Shadlen and Newsome, 2001*). This approach means that subjects were forced to integrate across the entire field of moving dots to establish the motion direction; tracking a single dot is not reliable because it only reappears on every third frame and does not necessarily move coherently for more than two frames.

## Conditions

To understand whether participants can flexibly adapt their integration kernel, we tested the continuous evidence integration task under different conditions following a within-subject 2 × 2 design. In different 5 min blocks, participants were told that they would either have long (5 s) or short (3 s) response periods, and either frequent (baseline period range 3–8 s) or rare (baseline period range 5–40 s) response periods. Subjects received extensive training (see below) so that these environmental statistics were well learnt prior to the experimental session. During the experiment, they were cued as to which condition they were currently in, by (i) displaying in text at the beginning of each block (e.g. 'response periods are LONG and FREQUENT'); and (ii) having a different shape of fixation point (triangle, square, circle, star) for each of the four blocks. This meant that there was no inference nor memory required from the subjects to know which condition they were currently in.

## Training

Our training protocol was designed to overtrain participants to reach a high level of performance on the task and to minimise learning effects during the main testing session. Our training taught participants about the structure of the long/short and rare/frequent trial periods, how to discriminate such trial periods from background noise fluctuations, and crucially incentivised participants to maximise their overall reward rate.

Training consisted of a sequence of different tasks that incrementally trained participants on the random dot motion stimulus and the continuous nature of the task. First, participants were introduced to the conventional RDM task based on discrete trials (*Shadlen and Newsome, 2001*), initially with very strong motion coherences (–0.9, 0.9), and then progressively with motion coherences resembling those found in the main experiment [-0.5,–0.4, –0.3, 0.3, 0.4, 0.5].

Next, participants completed intermediate task versions that still consisted of discrete trials, but had features of the continuous task: in particular, noise fluctuations was superimposed on the mean motion coherence, as in the final continuous task version coherences would fluctuate throughout the trial. This meant that participants had to estimate the average motion direction across the entire trial, but respond before the trial ended. Trial lengths were 5 s (length of long response periods) or 3 s (length of short response periods) and the fixation point shrank over the course of the trial so that participants would have an idea of how much time they had to respond. Trials with a mean coherence of 0% were also included, to simulate baseline periods of the continuous task that participants had to contrast to the other trials; participants had to suppress a response in these trials.

Then, participants moved on to the continuous version of the task, but they were first trained on a paradigm with higher mean coherences during response periods; in addition, the fixation dot would change its colour to white to indicate the onset of response periods. Gradually, all conditions of the experiment and the final mean coherence levels of signal periods were introduced, and a change of the fixation dot colour to white was disabled. At this point, when the paradigm was the same as in the full task, they had an extended period of practice on the task across all four conditions. Note that during this time the colours of the fixation dot feedback were the same as in the main experiment, and participants were instructed about the meaning of these dots.

We progressed participants through the different versions of training by checking psychometric functions to establish that they had fully learnt each stage before progressing onto the next stage of the task. Participants had to perform at 80% correct or higher on discrete trials to move on to the continuous tasks, and performance in the continuous task was checked qualitatively by plotting the stimulus stream and responses after each block. Verbal feedback was given to participants based on their performance, which also helped participants to improve their behaviour during training. If, in the latter parts of training, participants still missed more than half of the response periods, they were excluded from the subsequent EEG session. Participants (n = 3) who failed to progress from the training session were paid for that session and did not progress on to the EEG session.

After participants completed training successfully, they participated in the EEG testing session not more than 1 wk later. In this session, participants first performed a 'reminder' where they practised one run of the full task for 20 min (all four conditions, presented for 5 min each, in randomised order); this was performed while the experimenter put on the EEG cap. Then, while EEG data was collected, they completed 5–6 task 'runs', each lasting 20 min. Each run consisted of all four conditions in randomised order.

## Data collection

We tested 33 participants (13 male). Of those, three were unable to learn the continuous task and did not progress beyond training. One participant was excluded for falling asleep during the EEG session. Another five were excluded from the analysis due to technical issues matching the continuous EEG with the stimulus stream and/or issues with EEG data quality after pre-processing. This means 24 subjects were included in the analysis. Each subject completed six runs, except for one subject who completed only five runs. For the control experiment with superimposed vertical motion (*Figure 8*), a further six participants were tested (four males). All participants were aged 18–40, had normal or corrected-to-normal vision, and gave written consent prior to taking part in the study. The study was approved by the University of Oxford local ethics committee (CUREC R60298).

## Behavioural analysis

### Detection rate/reaction times

We calculated correct detection rate (*Figure 2b*) as the proportion of response periods in which correct responses were made. We calculated this separately for each run within each level of motion coherence (collapsing across leftward/rightward correct responses), and then averaged across the six runs, to obtain three values (0.3, 0.4, and 0.5 motion coherence) for each of the four conditions per subject. We performed a similar analysis on reaction times for these correct responses (*Figure 2c*), but here we excluded responses in LONG conditions that exceeded 3.5 s, such that the average response time could be directly compared between LONG and SHORT conditions. (We note that due to noise and the associated uncertainty concerning response period onset, participants are incentivised to respond as quickly as possible whenever they thought they were in a response period, as delaying responses would lead to 'missed trials'.) We analysed the effects of coherence, response period length and response period frequency on detection rate and reaction time across the 24 participants using a three-way repeated-measures ANOVA.

### Integration kernels

We calculated integration kernels by averaging the presented motion coherence for 5 s preceding every buttonpress (having first multiplied this by –1 for leftward buttonpresses, so that left and right responses can be averaged together). We did this separately for false alarms (*Figure 3b*) and for correct responses (*Figure 2d*). We excluded correct responses in LONG conditions that exceeded 3.5 s for similar reasons as outlined above. We also note that the integration kernel in this period includes a mixture of 'signal' (shift in mean coherence) plus noise, whereas the integration kernel from false alarms is driven by noise alone. This explains why the segment of the integration kernel that reflects non-decision time (i.e. immediately prior to buttonpress) returns close to 0 in *Figure 3b* but is closer to 0.5 in *Figure 2c*, and also why the false alarm integration kernel is more clearly an exponential decay function.

We fit an exponential decay model to the empirical integration kernel from false alarm responses:

$$k\left(t\right) = Ae^{\frac{-t}{\tau}}$$

where $k(t)$ is the height of the integration kernel $t$ seconds before its peak; $A$ is the peak amplitude of the integration kernel (in units that denote the fraction of dots moving towards the chosen response direction), and $\tau$ is the decay time constant (in units of seconds). To fit the exponential decay function, we first found the peak of the empirical integration function (using *max* in MATLAB), and set this timepoint to $t = 0$ in the equation above. We then fit $A$ and $\tau$ to the empirical integration kernel for all timepoints up to and including $t = 0$ using *fminsearch* in MATLAB using a least-squares cost function between the fitted model and data with an L2 regularisation term that penalised large values of either $A$ or $\tau$ ($\lambda = 0.01$). To calculate the quality of the model fit, we calculated $R^2$ for this function:

$$R^2 = 1 - \frac{RSS}{TSS}$$

with RSS being the residual sum of squares after model fitting and TSS being the total sum of squares.

### False alarm rates

To calculate false alarm rates (*Figure 3*), we counted the total number of responses made during baseline periods and divided this by the total amount of time where subjects could possibly have made a false alarm (i.e. total time spent in baseline periods). We repeated this separately for each of the four conditions within each participant.

### EEG acquisition

EEG data was collected at a sampling rate of 1000 Hz with Synamps amplifiers and Neuroscan data acquisition software (Compumedics) and 61 scalp electrodes following the 10–20 layout. Additionally, bipolar electrodes were placed below and above the right eye and on the temples to measure eyeblinks as well as horizontal and vertical eye movements (HEOG and VEOG channels). A ground electrode was attached to the left elbow bone. The EEG signal was referenced to the left mastoid but later re-referenced to the average of left and right mastoids. Impedances of electrodes were kept below 15 kΩ.

### EEG pre-processing

Data were pre-processed using spm12 (http://www.fil.ion.ucl.ac.uk/spm/; *Litvak et al., 2011*), the FieldTrip toolbox for EEG/MEG-analysis (http://fieldtriptoolbox.org; *Oostenveld et al., 2011*), and MATLAB (Version R2018b, The MathWorks, Inc, Natick, MA). Each session for each participant was pre-processed as continuous data. First, each session was downsampled to 100 Hz. Then, the data was rereferenced to the average of left and right mastoid electrodes and bandpass filtered the data between 0.1 Hz and 30 Hz using the function *spm_eeg_filter* with default settings (fifth-order Butterworth filter, passed in both directions). In a next step, we used signal space projection methods in SPM to perform eyeblink correction. The bipolarised VEOG channel was used to build a spatial confound topography of eye blinks to delineate ocular source components, Segments of 1000 ms around eye blink events in the VEOG channel were generated and averaged. Principal component analysis was then used to define the noise subspace of eyeblinks across all channels, and the first principal component was regressed out of the continuous EEG data (*Berg and Scherg, 1994*; *Hunt et al., 2012*). For each participant and session, the spatial confound map of the first component was visually checked to ensure it showed a typical eye blink topography before the regression was applied. The EEG data was further thresholded to remove artefacts that were ≥100 µV in a single channel by labelling a 500 ms window around the peak of the artefact, and removing these time windows when estimating the deconvolutional GLM.

### Deconvolutional GLM analysis

We used triggers sent to each jump in the noise stream to align the continuous EEG data with the continuous stream of sensory evidence (and other experimental events, such as buttonpresses). As the downsampled EEG was at the same sampling rate as the refresh rate as the display (100 Hz), we simply used the continuous stream of evidence presented on each frame of the experiment from then onwards. In addition to the five participants excluded due to technical issues with trigger recording and alignment (see 'Data collection' above), there was one further participant in our main EEG sample (n = 24) who had 3 out of 24 blocks missing due to technical issues; this participant was nevertheless taken forward into the main analysis with the remaining 21 recorded blocks.

We then constructed a design matrix *X* for the continuous EEG data, with 11 regressors in total:

$$EEG \sim jump_{event} + jump_{level} + jump_{|\Delta evidence|} + continuous_{|evidence|} + continuous_{evidence(signed)} +$$
$$response\ period\ onset_{event} + response\ period\ onset_{coherence} + buttonpress_{correct} + buttonpress_{false\ alarm} + (left-$$
$$right\ buttonpress)_{correct} + (left - right\ buttonpress)_{false\ alarm}$$

The 'buttonpress' regressors and the regressors with subscript 'event' are 'stick functions' (1 at the timepoint that they occurred, and 0 at all other timepoints). Other regressors are parametric modulators of these, except for the two continuous regressors which were valued at all timepoints of the experiment (reflecting the current motion onscreen, either absoluted [reported in the main text] or signed [not discussed]). In this article, we focus on responses to $jump_{event}$ (*Figure 6a*), $jump_{|\Delta evidence|}$ (*Figures 6b and 7a*, *Figure 7—figure supplement 1a*, *Figure 8*), $continuous_{|evidence|}$ (*Figures 6c and*

**Table 1.** (Average) explained variance between regressors of the convolutional general linear model (GLM) for baseline periods.

Columns and rows are the different regressors used to investigate baseline periods. Between each pair of regressors for the key continuous variables, the explained variance (squared correlation coefficient) was calculated to ensure that these regressors were not correlated with each other prior to estimating the GLM.

| $R^2$ | Jump | Jump level | Jump |Δevidence| | Continuous |evidence| |
|---|---|---|---|---|
| Jump | 1 | 0 | 0.01 | 0 |
| Jump level | 0 | 1 | 0.18 | 0.03 |
| Jump |Δevidence| | 0.01 | 0.18 | 1 | 0.01 |
| Continuous |evidence| | 0 | 0.03 | 0.01 | 1 |

*9*), and *buttonpress$_{correct}$* (*Figure 7b*, *Figure 7—figure supplement 1b*). For all of these except for the *buttonpress$_{correct}$* , we only estimate the EEG response during the baseline periods, when participants are still integrating evidence (as shown empirically in *Figure 3*), but the statistics of the stimulus stream across all four experimental conditions are matched. We calculated the correlation between the key regressors of interest (*Table 1*) to ensure that they were sufficiently decorrelated from one another to reliably obtain parameter estimates in the GLM.

To obtain the deconvolved response to each of these regressors, we time-expanded the design matrix into a large design matrix $X_{dc}$ (see *Ehinger and Dimigen, 2019* for a recent review; *Table 2*). We used simple 'staircasing' of the regressors to create this design matrix (as illustrated in *Figure 6*), rather than a time-Fourier basis set (*Litvak et al., 2013*) or time-Spline basis set (*Ehinger and Dimigen, 2019*); any of these approaches might be suitable for future studies. The number of timepoints for each of the regressors varied slightly between different regressors (e.g. we were primarily interested in activity *after* stimulus changes but *before* buttonpresses; the number of pre- and post-event lags reflected this). We then estimated parameter estimates for the deconvolved regressor at each sensor

**Table 2.** Design matrix.

This table describes the size of the design matrix assuming a sampling frequency of the electroencephalographic (EEG) signals of 100 Hz. For each regressor the number of lags pre- and post event and the total number of rows this regressor covers in the design matrix are described. The same number of lags was applied to vertical motion regressors for the control study.

| Regressor | Pre-event time in time-expanded design matrix (ms) | Post-event time in time-expanded design matrix (ms) | Total rows in the time-expanded design matrix |
|---|---|---|---|
| Jump event (stick function) | 1000 | 1500 | 251 |
| Jump level (|evidence| at each jump event) | 1000 | 1500 | 251 |
| Jump |Δevidence| (at each jump event) | 1500 | 1500 | 301 |
| Continuous |evidence| | 1500 | 1500 | 301 |
| Continuous (signed) evidence | 1500 | 1500 | 301 |
| Correct buttonpresses (stick function) | 5000 | 3500 | 851 |
| Correct buttonpresses (+1 for right, –1 for left) | 5000 | 3500 | 851 |
| False alarm buttonpresses (stick function) | 5000 | 3500 | 851 |
| False alarm buttonpresses (+1 for right, –1 for left) | 5000 | 3500 | 851 |
| Onset of response period (stick function) | 500 | 8000 | 851 |
| Response period |coherence| of response period (stick function) | 500 | 8000 | 851 |

for each subject with ordinary least squares (using the method of *Courrieu, 2008* to facilitate fast computation of the pseudoinverse of the design matrix).

## Permutation test for convolutional GLM analysis

To test for significant differences between deconvolved EEG responses for LONG versus SHORT response periods, and for RARE versus FREQUENT response periods, we performed a non-parametric paired *t*-test controlling for multiple comparisons across time, using the FieldTrip function *ft_timelockstatistics* (*Maris and Oostenveld, 2007*). We first selected electrodes and time windows of interest based upon the average response to the key regressors across all four conditions (see *Figure 6*); we note that because this selection vector is orthogonal to the difference between conditions (and the number of observations are matched between conditions), then it provides an unbiased method for selecting a window of interest (*Kriegeskorte et al., 2009*). In practice, this meant that the cluster-based permutation test was performed on an average of three centroparietal electrodes (CP1, CP2, CPz) and a time window from 0 to 800 ms post-event for jump-locked events (*Figure 7a*, *Figure 7— figure supplement 1a*, *Figure 8*), and an average of six centroparietal and central electrodes (C1, C2, Cz, CP1, CP2, CPz) and a time window from 2000 ms to 0 ms pre-event for buttonpress events (*Figure 7b*, *Figure 7—figure supplement 1b*). In total, 1000 permutations were generated with the Monte Carlo method, and clusters were selected based on a *T*-statistic threshold of 2.07 for initial cluster formation (except for $|jump_{|Devidence|}|$, where a slightly lower threshold of $T > 1.80$ was used), and an alpha of 0.05 (two-tailed) was then used for significance detection of clusters, corrected for multiple comparisons across time.

For the behavioural-neural correlations in *Figure 9*, we first temporally smoothed single subject betas with a Gaussian kernel with 75 ms FWHM (to further improve single subject SNR), and then calculated the Spearman's correlation at each timepoint between the estimated betas for the $continuous_{|evidence|}$ regressor and the $\tau$ parameters fit to the empirical evidence integration kernels for false alarms. We did this separately for the four conditions, providing four separate tests of the same behavioural-neural correlation (we note that these tests are independent in the sense that they consist of separate data for each correlation, but not in the sense that different participants were used to generate the data). In *Figure 9b*, we report the behavioural-neural correlation for the time window 420–750 ms after the evidence.

## Acknowledgements

We thank Neb Jojanovic for help in collecting the control experiment with simultaneous vertical and horizontal motion. MR is supported by a PhD studentship from the Wellcome Trust (109064/Z/15/Z). JOR is supported by a Career Development Fellowship from the Medical Research Council (MR/L019639/1) and an MRC Transition Support award (MR/T031344/1). LTH is supported by a Sir Henry Dale Fellowship from the Royal Society and the Wellcome Trust (208789/Z/17/Z). The Wellcome Centre for Integrative Neuroimaging is supported by core funding from the Wellcome Trust (203139/Z/16/Z). This research was funded in whole, or in part, by the Wellcome Trust. For the purpose of Open Access, the authors have applied a CC-BY public copyright licence to any Author Accepted Manuscript version arising from this submission.

## Additional information

### Funding

| Funder | Grant reference number | Author |
| --- | --- | --- |
| Wellcome Trust | 109064/Z/15/Z | Maria Ruesseler |
| Medical Research Council | MR/L019639/1 | Jill O'Reilly |
| Medical Research Council | MR/T031344/1 | Jill O'Reilly |
| Wellcome Trust | 208789/Z/17/Z | Lilian Aline Weber Laurence Tudor Hunt |

| Funder | Grant reference number | Author |
|--------|------------------------|--------|
| Wellcome Trust | 203139/Z/16/Z | Maria Ruesseler<br>Lilian Aline Weber<br>Tom Rhys Marshall<br>Jill O'Reilly<br>Laurence Tudor Hunt |

The funders had no role in study design, data collection and interpretation, or the decision to submit the work for publication. For the purpose of Open Access, the authors have applied a CC BY public copyright license to any Author Accepted Manuscript version arising from this submission.

## Author contributions

Maria Ruesseler, Conceptualization, Resources, Data curation, Software, Formal analysis, Funding acquisition, Investigation, Visualization, Methodology, Writing – original draft, Writing – review and editing; Lilian Aline Weber, Resources, Data curation, Software, Formal analysis, Validation, Visualization, Writing – review and editing; Tom Rhys Marshall, Conceptualization, Formal analysis, Supervision, Investigation, Methodology, Writing – review and editing; Jill O'Reilly, Conceptualization, Formal analysis, Supervision, Methodology, Project administration, Writing – review and editing; Laurence Tudor Hunt, Conceptualization, Resources, Data curation, Software, Formal analysis, Supervision, Funding acquisition, Investigation, Visualization, Methodology, Writing – original draft, Project administration

## Author ORCIDs

Lilian Aline Weber  http://orcid.org/0000-0001-9727-9623
Laurence Tudor Hunt  http://orcid.org/0000-0002-8393-8533

## Ethics

Human subjects: All participants gave written consent prior to taking part in the study. The study was approved by the University of Oxford local ethics committee (CUREC R60298).

## Decision letter and Author response

Decision letter https://doi.org/10.7554/eLife.82823.sa1
Author response https://doi.org/10.7554/eLife.82823.sa2

# Additional files

## Supplementary files

• MDAR checklist

## Data availability

Code repositories are available at our lab GitHub site for recreating the experimental paradigm within Psychtoolbox (https://github.com/CCNHuntLab/continuous-rdm-task, copy archived at *Hunt, 2022a*) and for analysis of both behavioural and EEG data in MATLAB (https://github.com/CCNHuntLab/ruesseler-eeg-analysis, copy archived at *Hunt, 2022b*). A resource containing both raw and pre-processed anonymised EEG and behavioural data has been uploaded to DataDryad, and is available at https://doi.org/10.5061/dryad.02v6wwq6b.

The following dataset was generated:

| Author(s) | Year | Dataset title | Dataset URL | Database and Identifier |
|-----------|------|---------------|-------------|-------------------------|
| Ruesseler M, Weber LA, Marshall TR, O'Reilly J, Hunt LT | 2023 | Decision-making in dynamic, continuously evolving environments: Quantifying the flexibility of human choice | https://dx.doi.org/10.5061/dryad.02v6wwq6b | Dryad Digital Repository, 10.5061/dryad.02v6wwq6b |

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
