## [Editor Report]

This important study by Ruesseler, Weber and colleagues employs psychophysical kernels and EEG reverse correlation methods to identify the decision process adjustments used to account for variations in target frequency and duration in a task in which targets emerge periodically within a continuous stimulus stream. The paper provides solid evidence for the role of leak and threshold adjustments. The paper will be of interest to researchers studying mathematical models and neurophysiological correlates of decision making and more broadly authors with an interest in the application of reverse correlation techniques for neural signal analysis.

---

## [Decision Letter]

**Decision letter after peer review:**

Thank you for submitting your article "Decision-making in dynamic, continuously evolving environments: quantifying the flexibility of human choice" for consideration by *eLife*. Your article has been reviewed by 3 peer reviewers, including Redmond G O'Connell as the Reviewing Editor and Reviewer #1, and the evaluation has been overseen by Michael Frank as the Senior Editor. The following individuals involved in the review of your submission have agreed to reveal their identity: Charline Tessereau (Reviewer #3).

Essential revisions:

1) The conclusions drawn from the integration kernel analyses appear difficult to reconcile with key features of the behavioural data. For example, the authors argue that the kernel analyses point to a difference in leak between the frequent and rare conditions but no corresponding change in false alarm rate is observed. Similarly, where the authors conclude from kernel analyses that participants did not alter their decision bounds between frequent and rare conditions the motor preparation ERP signal suggests that there is a bound adjustment. A key issue here is that it is not clear that integration kernels can be used to conclusively discriminate between a leak, bound or drift rate adjustment as each of these could conceivably impact the kernel. In order to make sense of the dataset, the authors should consider fitting alternative model variants to the data or, at minimum, simulating the impact of different decision process parameter adjustments on the integration kernel in order to determine whether their conclusions are warranted.

2) The reviewers raise questions regarding the nature of the decision required by the particular task implemented by the authors. The authors' approach relies on the assumption that participants are simply integrating coherent motion as they might for a standard variant of the random dot motion task. However on this task, targets were actually defined not by the presence of coherent motion but by a criterion duration of coherent motion, during which time the frame-to-frame coherent motion variance diminished. Can the authors rule out the possibility that decisions were partly or entirely based on detecting changes in stimulus variance and or coherent motion duration? If not, can it still be assumed that the rare and frequent targets were matched in terms of the ease with which they could be distinguished from the intervening noise intervals?

*Reviewer #1 (Recommendations for the authors):*

My principle suggestion would be that the authors consider running simulations of alternative models to see if they can reproduce the current behavioural trends. Drift rate adjustment to Rare vs Frequent target conditions is an obvious candidate.

In general, I found it hard to marry up all the different behavioural/kernel/EEG findings. I recommend the authors examine whether leak adjustment really provides a comprehensive account of their data or if additional/alternative parameter adjustments must be invoked to reconcile their data.

*Reviewer #2 (Recommendations for the authors):*

1) While I don't think it is problematic, the authors should acknowledge (even if just in methods) that subjects might be able to use the stimulus variance to identify response periods. Also, strictly speaking, the description of the noise as following a normal distribution cannot be correct because it is limited to the range of [-1,1]. I presume this also explains why the authors reduced the standard deviation in the response periods – to avoid hitting those edges too often.

2) Figure 1d is very challenging to interpret. I appreciate that the authors fully acknowledge that it involves signal and noise. However, I think this introduces so many confounds to that analysis that I'm not sure you can say much with confidence from it. Most obviously, the dynamics will reflect the RT differences and the associated shift in the onset of the signal contribution. Also, because it seems to be based on all coherences together, the signal contribution will critically depend on the relative proportion of each coherence in each condition. I'm not sure there is any way to disentangle all of that.

3) I wasn't quite sure how false alarm rate was calculated in Figure 3. Conditions differ in the total time for every run that subjects were in the baseline state where they could have a false alarm. Is the denominator of this rate calculation total task time or total time where subjects could possibly have a false alarm. I think the latter is the most useful comparison. I would guess the authors did that, but I couldn't find the description.

4) It would be useful to have a figure showing individual subject CPP effects shown in aggregate in Figure 5? A scatter plot comparing effect size for FREQUENT and RARE would be a good way to illustrate that.

5) In the discussion, the authors state that initial simulations suggest that a range of decay time constants could be optimal. This may provide another explanation for the lack of change in time constant in different conditions. If a range are possibly optimal for any given condition, perhaps the subjects can remain close to optimal in this task without changing their decay time constant between conditions.

6) I noticed two typos/formatting errors in the methods. In the subscripts for the regression, a "D" was shown for the "δ" symbol. Also, the words "regressed out" are duplicated in EEG pre-processing section.

*Reviewer #3 (Recommendations for the authors):*

General notes:

– Figure captions should describe what is being shown, with a link to the method and/or supplementary material for lengthy computations.

– Throughout the text, the authors sometimes refer to 'net motion', 'coherence', and 'evidence'. I would make sure to use the same terminology.

– Some analyses are missing, e.g. the proportion of long answers that have been ignored (that could indicate a level of engagement in the task).

Detailed comments/suggestions:

Task design:

– Methods: are subjects instructed about what the dots mean in advance? How long do the dots stay colored for during feedback?

– How much do participants get in total? The total bar is 30 points and they get only 50 cts to have 15 points which are after 5 good responses without error on average. Are the feedback distracting the participants? Do the authors think that the money aspect here provides a substantial motivation for participants?

– Figure 1c:" The resulting evidence integration kernel is well described by an exponential decay function, whose decay time constant in seconds is controlled by the free parameter tau ": As formulated, the caption hints that the figure analyses how well described is the kernel. As I understand, the exponential kernel is a choice that is imposed by the authors, and how well it fits the coherence level before a decision is not addressed in the paper. I would reformulate and clearly state that the authors define an exponential kernel from the past coherence levels. More details on the definition of this kernel are required, and statistics about how well it fits the data would enable stronger claims of its use throughout the text. In particular, the average coherence can take 3 values (30, 40, or 50), are the authors averaging for all coherence levels together? Should A be correlated with those values? What is the effect of the mean coherence level during response periods?

P7, Figure 2:

– b and c: is that the mean or median (for b) and the error bar the std/95expectiles (for c) of participants' detection rate/RT? Is the RT tau? I know both variables are defined in the methods section but they should also be explained here or a link should be made to the method section so that the reader can know what they are looking at.

– d: is the integration kernel here referring to the exponential, or simply the average signal before the decision, averaged across all participants?

– d again: why is the maximum coherence 0.5 here? is that only for switches from 0 to 0.5 coherence? What happens for switches from 0 to 30 coherence?

– d again: the authors observe no effect on the response window duration on the response kernel, but have applied a threshold on the duration of the response on the long response period in order to make this plot. What happens if the authors include all response lengths? (also applies to the discussion p17 and 18).

P 8:

– "This provides further evidence that participants were overall more conservative in their responses in LONG conditions than SHORT" can the authors define what conservative means?

– It is not surprising that the integration kernels are not affected by the length of the response period if the long responses are not considered.

– I think the kernel analysis during baseline is necessary to show that participants are not randomly making motor movements, however, the authors could describe in a bit more detail the difference between a correct response and a false alarm: how are the integration kernels different in the baseline vs response period? (the kernels for false alarms seem a bit shorter?)

– When do false alarms occur compared to the response periods? Are they regularly after the end of a response period? What is the pattern of the coherence that causes those false alarms? Are they always when the frame update is 1s, for example?

Figure 3:

a) the authors mention significance but do not provide p values. Why is it consistent with having a more cautious response threshold? I'd favor the reverse interpretation that short response periods induce more confusion between signal and noise.

b) is that the averaged signal before false alarms? Do they always occur when the mean average coherence reaches 0.5? Why is the maximum value 0.5?

c) explain what the points show and how these are computed.

P.11

(ii) To me, it is not clear that the behavioral evidence from 'false alarms' indicate that participants are still integrating sensory evidence – a stricter definition and existence of the kernel (how well it fits for every response) could help support this statement

– Is 'net motion' coherence here?

– On the same page – the regressor 'absoluted sensory evidence' has not been named like this before.

P. 12:

– Why is it consistent with the behavioral finding of lower detection rate and slower RT in rare conditions?

– I understand the analysis is done regardless of the response. What about missed response periods/responses during the baseline?

Figure 4:

– The caption is incomplete, what does the right-hand side show? Is it one weight for one regressor across time for one example electrode?

– What is the goodness of fit of the regression?

P13 – Figure 5:

– What are the lines showing? What is the black thick line on top?

– P.14 To test whether these signals are decision-related, the authors could also compare the results of the regression when false alarms are made and when the response period is missed.

– Figure 6: has the effect of previous exposure been controlled for the vertical motion? Could the authors discuss whether this would have an impact on the CPP.

– P 15: could the authors give more details on the behavioral correlate and how is it computed.

– "this negative-going component was larger in amplitude (i.e. more negative) in participants who would integrate sensory evidence over longer durations (i.e. had a higher value of τ).": could the authors elaborate on how to interpret this finding with regard to existing literature?

– Figure 7 P 16 spearman correlation: can the authors describe the top plot? what are the lines showing? if that's the mean it should be mentioned and the standard deviation should be shown. Can the authors describe the bottom plots? why are the authors using log(tau) for the x-axis?

Discussion:

- p 17: 'The behavior is well described by a leaky evidence accumulator' the exponential kernel is a reverse correlation that the authors have done themselves, but they have not shown causality nor statistical analysis on what those kernels mean in relationship with the behavior.

Methods:

– In the regressor table, what is 'prediction error' regressor?

---

## [Author Response]

Essential revisions:1) The conclusions drawn from the integration kernel analyses appear difficult to reconcile with key features of the behavioural data. For example, the authors argue that the kernel analyses point to a difference in leak between the frequent and rare conditions but no corresponding change in false alarm rate is observed. Similarly, where the authors conclude from kernel analyses that participants did not alter their decision bounds between frequent and rare conditions the motor preparation ERP signal suggests that there is a bound adjustment. A key issue here is that it is not clear that integration kernels can be used to conclusively discriminate between a leak, bound or drift rate adjustment as each of these could conceivably impact the kernel. In order to make sense of the dataset, the authors should consider fitting alternative model variants to the data or, at minimum, simulating the impact of different decision process parameter adjustments on the integration kernel in order to determine whether their conclusions are warranted.

We thank the reviewers and editor for this suggestion – it has been a very important comment for reshaping our paper and the interpretation of several of the behavioural findings. In short, we agree with the reviewers that the integration kernels do not conclusively point to a change in leak, and in fact they are equally (if not more) likely to reflect a change in decision threshold. We have rewritten and extended parts of the paper substantially to reflect this change.

With respect to model fitting, we should first mention that we have found it challenging to *directly* fit models to the behavioural data. This is because, unlike for trial-based paradigms, there are not straightforward solutions to parameter fitting for continuous paradigms such as ours at present. We have quite extensively explored use of toolboxes such as the generalized drift diffusion model to study our data (Shinn et al., 2020), but it is difficult to adapt these toolboxes to settings in which responses can be emitted at any point in a continuous motion stream. We note that there are ongoing efforts that address model fitting using similar continuous paradigms (Geuzebroek et al., 2022), but these depend upon using a cost function that makes use of reaction time quantiles after target onset as well as for false alarms. A key difference in our paradigm is that the injection of continuous, structured noise into the sensory evidence complicates the interpretation of RT quantiles, and particularly false alarms, as the noise affects when false alarms will occur (and also the starting point of the accumulator at target onset). A more appropriate approach to fitting subjects’ behaviour in our task would take account of the impact of the structured noise on when the accumulator hits the decision threshold. This is something that we hope to pursue further in future work but were unable to successfully implement for the current paper.

We have therefore instead pursued the second suggestion in this comment (“simulating the impact of different decision process parameter adjustments on the integration kernel in order to determine whether their conclusions are warranted”). We have inserted a new section into the paper, entitled “Computational modelling of leaky evidence accumulation” (p.11 onwards in revised manuscript), with three additional figures to present these results.

We have simulated a leaky evidence accumulation model (Ornstein-Uhlenbeck process) that takes the actual presented stimulus stream (signal+structured noise) as its input, and have then simulated the impact of variations in leak and threshold on the model’s behaviour. (We adopted reviewer 2’s suggestion of discussing changes in threshold rather than reviewer 1’s suggestion of changes in gain; we discuss this further below). We present two sets of analyses from these simulations.

Our first set of model analyses explores how variation across leak and threshold parameters affects low-level behavioural features of the data: correct responses, missed responses, false alarm rates and overall reward obtained. The full exploration of these effects is shown in new Figure 4 —figure supplement 1 in the revised paper, and associated new main Figure 4.

This analysis highlights how the optimal strategy for solving the task (in terms of maximising reward gained) differs between the four conditions of interest, and in ways that we would have expected. For example, a higher decision threshold is beneficial when response periods are RARE rather than FREQUENT. The analysis also highlights an important point about the *relationship between* these parameters. In particular, there tends to be a ‘ridge’ in parameter space where several different model parameterisations perform similarly well in terms of maximising total points gained. This ‘ridge’ of peak performance falls along a line where leak (λ) is decreasing and decision threshold is increasing. In other words, a less ‘leaky’ integration model retains more evidence in the accumulator; this can be accommodated by an increase in threshold to perform similarly on the task.

We have included a new Figure 4 that illustrates these points:

We note that this may explain why we observe greater levels of *between-subject* variation than *between-condition* variation in task performance. There are a wide number of combinations of leak/threshold parameters that can perform well on the task (see new Figure 4 —figure supplement 1 for further detail), and the difference between conditions is comparably small. So, one possibility is that when learning the task, participants may perform gradient ascent on this parameter space to maximise the reward gained. They may all be able to perform well on the task, but in doing so they may reach different locations on this ‘ridge’. Between-condition variation could then account for a small movement of these parameters within each participant. As a result, participants may show considerable variation in integration kernels (Figure 3 —figure supplement 1), and closely related changes in neural measures of evidence integration (Figure 9), without showing much variation in reward obtained.

We suggested that this might be a possibility in our previous discussion, and we have now amended this to include the new findings from the modelling:

“An alternative hypothesis is that the individual variability we observe may be a consequence of the prior expectations that our participants have about the overall task structure, combined with learning over the course of training. One result in support of this hypothesis comes from the modelling shown in Figure 4 Not only does the result in Figure 4a show that behaviour should be adapted across different conditions, but it also shows that different individuals might potentially achieve similar performance by ending up at very different locations in this parameter space. This could in turn explain why between-subject variability in these kernels exceeded between-condition variability (Figure 3c; Figure 3 —figure supplement 1). During training, different participants could have optimised their parameters to maximise points gained, but in doing so ended up at different locations on the ‘ridge’ of parameters shown in Figure 4a. To adapt behaviour between conditions, they may have then made a small adjustment in these parameters to optimise performance for each environment…”

We also note that this analysis suggests that changes in one parameter are unlikely to occur truly independently of changes in other parameters. Optimising performance may involve a joint adjustment in leak and threshold, and conversely it may be difficult to uniquely determine the parameters from behaviour in this version of the task. In future work, it may be worth considering reparameterizations of the model that allow for true independence in parameter space, or designing experiments in which only small area of parameter space is optimal rather than a ‘ridge’ of parameter space. We have also mentioned these points in our revised discussion:

“Further work will be needed to distinguish these explanations of between subject variability in integration kernels, and to test competing models of participant behaviour. Although the Ornstein-Uhlenbeck process that we use is an appropriate and widely used model of the task, alternative models might also consider a dynamically changing threshold as a function of progress through the inter-trial interval (Geuzebroek et al., 2022); or consider tracking the mean and variance of the stimulus over time, rather than just the mean (Bill et al., 2022). In the current work, we also did not directly fit parameters of the Ornstein-Uhlenbeck process to participant behaviour. Although progress has recently been made in model fitting for decision making in continuous decision-making paradigms (Geuzebroek et al., 2022), a key feature of our paradigm is that many responses result from the structured noise that we inject into the sensory evidence stream, which complicates the use of aggregate measures such as reaction time quantiles for model fitting. Model estimation could potentially be improved by having continuous behavioural output, as recently demonstrated in tracking paradigms (Huk et al., 2018; Straub and Rothkopf, 2021).”

With this comment in mind, we turn to our second set of model analyses, which address the reviewers’ comments more directly. These assess the effects of manipulating leak and decision bound on task performance, and in particular on the integration kernels in our task. (See response to reviewer 1, below, for why we decided to principally focus on these two parameters rather than also considering changes in gain).

Our principal finding from this analysis was the (counterintuitive) finding that variation in decision threshold*,* rather than variation in leak, was the principal controller of the decay time constant t of the ‘evidence integration’ kernels that we obtain behaviourally. We demonstrate this in a new Figure 5 which shows the effect of independently manipulating these three parameters on the recovered evidence integration kernels from the model.

This finding was a surprise to us (but perhaps not the reviewers, given their comments above). After all, it is the leak (not the threshold) that affects the internal decay time constant of the accumulator, so why does the threshold (not the leak) primarily affect the recovered integration kernels in this way?

The answer lies in the fact that the data that enters the integration kernels is itself a ‘thresholded’ process. Only when the model reaches threshold and a response is emitted does the evidence preceding that response enter the integration kernel analysis. As such, an elevated threshold will require a more sustained (over time) level of sensory evidence to emit a response, and so give rise to the effects on integration kernels shown above.

We thank the reviewers for clarifying this important point in our analysis. We hope that the additional modelling section that we have added to the paper (which we do not reproduce in full here, due to length) fairly reflects the results of this attempt at modelling our data. We think that it adds substantially to the paper, and the resulting conclusion (the key result is likely a change in threshold, rather than/in addition to any change leak) is indeed more consistent with other features of the data. For example, the greater negative-going potential immediately prior to a button press (Figure 7b) is consistent with a higher threshold during rare vs. frequent trials; the reduced detection rate and increased reaction times seen in these trials is also consistent (Figure 2b/c).

Nonetheless, completely disentangling the effects of threshold and leak is a thorny issue, as adaptations in one parameter can be compensated by a change in the other parameter, as shown in new Figure 4 —figure supplement 2 in the modelling section. To *truly* derive integration kernels in which the leak of evidence is more directly accessible to the researcher, we suggest that it may be necessary to design tasks in which the response outputs are not binary when a threshold is reached, but instead where the subject continuously reports their current estimates of the decision variable (Huk et al., 2018; Straub and Rothkopf, 2021). This problem is in fact directly analogous to a previous set of ideas in reinforcement learning, in which it was established that the learning rate (»leak) and softmax (»threshold) parameters are surprisingly difficult to disambiguate from one another in binary choice tasks (Nassar and Gold, 2013), and it was instead suggested to design tasks where the learning rate was reported directly (Nassar et al., 2012; O’Reilly et al., 2013).

2) The reviewers raise questions regarding the nature of the decision required by the particular task implemented by the authors. The authors' approach relies on the assumption that participants are simply integrating coherent motion as they might for a standard variant of the random dot motion task. However on this task, targets were actually defined not by the presence of coherent motion but by a criterion duration of coherent motion, during which time the frame-to-frame coherent motion variance diminished. Can the authors rule out the possibility that decisions were partly or entirely based on detecting changes in stimulus variance and or coherent motion duration? If not, can it still be assumed that the rare and frequent targets were matched in terms of the ease with which they could be distinguished from the intervening noise intervals?

We thank the reviewers for raising this important point. As reviewer #2 correctly infers below, we reduced the variance during the response periods to avoid hitting the edges of the normal distribution too often (as the distribution is truncated at [-1,+1]). We have now made this more explicit in the methods section:

“The level of motion coherence at each step was sampled randomly from a normal distribution. The mean of this normal distribution depends on whether the step occurred during baseline or a response period. During a baseline period the mean of the normal distribution is 0. That means it is equally likely that negative or positive coherences are drawn. During response periods, the mean of the normal distribution was sampled uniformly from the set [-0.5, -0.4, -0.3, 0.3, 0.4, 0.5]. Any samples that exceeded 100% motion were set to be [+1,-1]. To limit the number of times this occurred, we set the standard deviation of the distribution to 0.3 for response periods and 0.5 for baseline periods. (We note that this could allow a strategy of tracking changes in the variance in the stimulus as well as the mean, something that we address in Figure 2 —figure supplement 1).”

We hope that this clarifies why the variance changed between response and baseline periods. We now consider whether participants might track changes in stimulus variance and how this might affect our results.

Let us address the final question from the reviewers’ comment first. The rare and frequent targets are indeed matched in terms of the ease with which they can be distinguished from the intervening noise intervals. To confirm this, we directly calculated the variance (across frames) of the motion coherence presented during baseline periods and response periods (until response) in all four conditions:

**Author response image 1. sa2fig1:** The average empirical standard deviation of the stimulus stream presented during each baseline period (‘baseline’) and response period (‘trial’), separated by each of the four conditions (F = frequent response periods, R = rare, L = long response periods, S = short). Data were averaged across all response/baseline periods within the stimuli presented to each participant (each dot = 1 participant). Note that the standard deviation shown here is the standard deviation of motion coherence across frames of sensory evidence. This is smaller than the standard deviation of the generative distribution of ‘step’-changes in the motion coherence (std = 0.5 for baseline and 0.3 for response periods), because motion coherence remains constant for a period after each ‘step’ occurs.

The empirical standard deviation of the stimulus streams is well matched across the four conditions. As such, rare and frequent trials are equivalently difficult to distinguish from noise based on their variance. This means that any effects that we see across the four conditions are unlikely to be due to differences in low-level stimulus statistics.

However, the stimulus does substantially reduce its variance during response periods, as was part of the stimulus generation procedure. The reviewers therefore raise the important question about whether participants could use a strategy of variance tracking across time.

To explore this, we first considered an analogous analysis to the ‘integration kernels’ approach that we previously used to analyse how participants tracked changes in the mean in the experiment. We calculated a local running estimate of the variance of the previous 200 frames (2 seconds) of sensory evidence at each timepoint in the experiment. We then epoched and averaged this local estimate of variance immediately prior to each ‘false alarm’ response.

Using this approach, we found that the average variance of the stimulus stream decreased prior to false alarms emitted by the participants. This implies that participants did partially use variance tracking as a strategy for solving the task:

**Author response image 2. sa2fig2:** Integration kernels for stimulus variance, timelocked to false alarms made by participants in each of the four conditions. Each line shows mean +/- s.e.m. (across participants) of the integration kernel for stimulus variance; in all four conditions, the reduction instimulus variance prior to a false alarm indicates that participants were likely performing stimulus detection in part using information about stimulus variance as well as stimulus mean (main Figures 2d, 3b).

To confirm these effects weren’t artifacts, we also considered whether these kernels might occur because of the truncation of the normal distribution at [-1, +1]. (I.e. as the mean of the stimulus increases prior to a response, the variance may also potentially decrease as more samples may have been drawn from truncated values). To rule this out, we repeated this analysis using simulated date from the Ornstein-Uhlenbeck model of evidence accumulation explicated above. This model only tracks the mean of the stimulus by definition, and not its variance:

**Author response image 3. sa2fig3:** The same analysis as in Reviewer Response Figure 2, but now performed on simulated data from the Ornstein-Uhlenbeck model rather than participant behaviour. The absence of any integration kernel confirms that the results in the previous figure are not artifactual.

The absence of any integration kernel here confirms that the analysis is not prone to artifacts. We conclude that participants’ decisions are therefore at least *partially* based on detecting changes in stimulus variance across time, as well as changes in stimulus mean.

Given these findings, we next considered the *relative* contributions of mean and variance to participants’ decision-making. We performed logistic mixed effects modelling of detection probability during response periods, including both the mean *and* variance of the stimulus as co-regressors. We show these new results in the main paper as a new Figure 2 —figure supplement 1.

Note that this is essentially an extension of the analysis originally reported in main figure 2b, where we observed main effects of mean coherence, response period frequency and length on detection probability. We therefore included response period frequency and length as coregressors, along with relevant interaction terms.

We have described these results in the text below:

“We also considered an alternative stimulus detection strategy, of changes in stimulus variance across time rather than changes in stimulus mean. This hypothesis relied upon the fact that response periods had smaller standard deviations in the gaussian noise distribution than baseline periods – a stimulus feature that we introduced to avoid excessive samples of ‘maximal’ (100%) motion coherence when the mean was non-zero. To test whether the variance of the stimulus might also affect participants’ detection, we performed a logistic mixed effects model on participants’ responses (Figure 2 —figure supplement 1). Detection probability was the dependent variable, and mean motion coherence, variance of motion coherence, response period frequency and length were independent variables, along with interaction terms. We found that stimulus variance during response periods did indeed impact detection probability; response periods with a higher variance in motion coherence were less likely to be detected. Crucially, however, the main effects of mean motion coherence, trial frequency and trial length (equivalent to the effects plotted in main Figure 2b) were left unaffected by the inclusion of this coregressor.”

Reviewer #1 (Recommendations for the authors):My principle suggestion would be that the authors consider running simulations of alternative models to see if they can reproduce the current behavioural trends. Drift rate adjustment to Rare vs Frequent target conditions is an obvious candidate.In general, I found it hard to marry up all the different behavioural/kernel/EEG findings. I recommend the authors examine whether leak adjustment really provides a comprehensive account of their data or if additional/alternative parameter adjustments must be invoked to reconcile their data.Reviewer #2 (Recommendations for the authors):1) While I don't think it is problematic, the authors should acknowledge (even if just in methods) that subjects might be able to use the stimulus variance to identify response periods. Also, strictly speaking, the description of the noise as following a normal distribution cannot be correct because it is limited to the range of [-1,1]. I presume this also explains why the authors reduced the standard deviation in the response periods – to avoid hitting those edges too often.

The reviewer is correct on both counts. We have now acknowledged this in the methods section, as suggested.

“The level of motion coherence at each step was sampled randomly from a normal distribution. The mean of this normal distribution depends on whether the step occurred during baseline or a response period. During a baseline period the mean of the normal distribution is 0. That means it is equally likely that negative or positive coherences are drawn. During response periods, the mean of the normal distribution was sampled uniformly from the set [-0.5, -0.4, -0.3, 0.3, 0.4, 0.5]. Any samples that exceeded 100% motion were set to be [+1,-1]. To limit the number of times this occurred, we set the standard deviation of the distribution to 0.3 for response periods and 0.5 for baseline periods. (We note that this could allow a strategy of tracking changes in the variance in the stimulus as well as the mean, something that we address in Figure 2 —figure supplement 1).”

2) Figure 1d is very challenging to interpret. I appreciate that the authors fully acknowledge that it involves signal and noise. However, I think this introduces so many confounds to that analysis that I'm not sure you can say much with confidence from it. Most obviously, the dynamics will reflect the RT differences and the associated shift in the onset of the signal contribution. Also, because it seems to be based on all coherences together, the signal contribution will critically depend on the relative proportion of each coherence in each condition. I'm not sure there is any way to disentangle all of that.

This is a fair criticism. Nevertheless, we would be keen to leave this analysis in – not because of the *presence* of the difference between FREQUENT and RARE trials, but because of the *absence* of any difference between LONG and SHORT trials. As mentioned in the text, our original hypothesis was that participants might integrate longer in LONG than in SHORT, and this appears not to be the case:

“Surprisingly, the sensory evidence integration properties (i.e. the ‘integration kernels’, calculated by averaging the signal prior to the decision) were not affected by length of response periods (Figure 2d). This ran contrary to our initial hypothesis that participants would integrate evidence for longer when response periods were LONG. We suggest that this may result from the manipulation of response period duration being relatively small (3s vs. 5s) compared to the manipulation of response period frequency….”

The reviewer could counter that we draw the same conclusions from our integration kernels from false alarms – so why include a potentially confounded analysis at all? A potential concern is that there are generally far fewer false alarms than correct responses, and so the absence of any difference in LONG vs. SHORT in figure 3b/Figure 3 —figure supplement 1a could potentially be due to lack of SNR. So, we prefer to keep in the response period analysis, just as further evidence that the kernels really appear the same on LONG vs. SHORT.

We have, however, added additional text to further clarify how the analysis has the potential to suffer from confounds, and crucially that the subsequent false alarm analyses (which show the same direction of effects) are not affected by these confounds:

“…We also note that the significant difference between FREQUENT and RARE trials in Figure 2d should not be over-interpreted, as it could be influenced by RT differences (Figure 2c) and the associated shift in the onset of the signal contribution, and/or the difference in average coherence detection across conditions (Figure 2b). Importantly, we control for these confounds below, by examining the integration kernels to false alarms (in the absence of changes in mean signal).”

3) I wasn't quite sure how false alarm rate was calculated in Figure 3. Conditions differ in the total time for every run that subjects were in the baseline state where they could have a false alarm. Is the denominator of this rate calculation total task time or total time where subjects could possibly have a false alarm. I think the latter is the most useful comparison. I would guess the authors did that, but I couldn't find the description.

The reviewer is correct, the denominator here is total time where subjects could possibly have a false alarm (i.e. total time in baseline periods). We have added the following clarification to the methods:

“False alarm rates. To calculate false alarm rates (main Figure 3), we counted the total number of responses made during baseline periods, and divided this by the total amount of time where subjects could possibly have made a false alarm (i.e. total time spent in baseline periods). We repeated this separately for each of the four conditions within each participant.”

4) It would be useful to have a figure showing individual subject CPP effects shown in aggregate in Figure 5? A scatter plot comparing effect size for FREQUENT and RARE would be a good way to illustrate that.

Thanks for the suggestion! We have now included such a plot in the manuscript. To also facilitate comparison of the effects between FREQUENT/RARE and SHORT/LONG, we decided that it was easiest to include this as an additional figure supplement (new Figure 7 —figure supplement 2).

5) In the discussion, the authors state that initial simulations suggest that a range of decay time constants could be optimal. This may provide another explanation for the lack of change in time constant in different conditions. If a range are possibly optimal for any given condition, perhaps the subjects can remain close to optimal in this task without changing their decay time constant between conditions.

Yes – as described above, we have now explored this much more fully by simulating the O-U model and showing the range of parameter values where subjects can remain close to optimal. Although there are clear differences between conditions in terms of the exact parameters that are optimal between conditions, there is also a ‘ridge’ in parameter space where performance is close to optimal (see new main Figure 4 and Figure 4 —figure supplement 1). This may explain why between-participant variability exceeds between-condition variability in task performance.

6) I noticed two typos/formatting errors in the methods. In the subscripts for the regression, a "D" was shown for the "δ" symbol. Also, the words "regressed out" are duplicated in EEG pre-processing section.

Thanks – corrected.

Reviewer #3 (Recommendations for the authors):General notes:– Figure captions should describe what is being shown, with a link to the method and/or supplementary material for lengthy computations.

We have gone through all the figure captions, and amended these/added links to the methods/supplementary material wherever appropriate.

– Throughout the text, the authors sometimes refer to 'net motion', 'coherence', and 'evidence'. I would make sure to use the same terminology.

Thanks for this comment. In general, we try to use ‘sensory evidence’ or ‘evidence’ whenever we are making a general point about decision making via evidence accumulation across time, but use ‘motion coherence’ when we are referring specifically to our experimental paradigm. So, for example, we discuss evidence accumulation in general terms in the introduction, so we only discuss ‘sensory evidence’ there, but we start using the term ‘motion coherence’ at the beginning of the Results section where we first introduce our random dot kinetogram task. We have now made this transition in terminology more explicit in the text:

“Subjects continuously monitored a stream of time-varying sensory evidence (hereafter referred to as ‘motion coherence’) for blocks of five minutes…”

– Some analyses are missing, e.g. the proportion of long answers that have been ignored (that could indicate a level of engagement in the task).

We are not quite sure what the reviewer is referring to here – but we suspect that they mean the proportion of ‘missed response periods’ (i.e. responses that should have been made, but were not)? If so, this is simply 1 minus the ‘correct detection rate’, which is plotted in figure 2b. This is plotted separately for the four conditions, and the different levels of motion coherence, and so the ‘proportion of long answers that have been ignored’ can be read off directly from this graph.

Detailed comments/suggestions:Task design:– Methods: are subjects instructed about what the dots mean in advance? How long do the dots stay colored for during feedback?

We have added to the introduction:

“Feedback was presented by changing the colour of the central fixation point for 500ms (Figure 1a), and they were trained on the meaning of these colours as part of extensive pre-experiment training (see methods).”

And to the methods:

“Note that during this time the colours of the fixation dot feedback were the same as in the main experiment, and participants were instructed about the meaning of these dots.”

– How much do participants get in total? The total bar is 30 points and they get only 50 cts to have 15 points which are after 5 good responses without error on average. Are the feedback distracting the participants? Do the authors think that the money aspect here provides a substantial motivation for participants?

The exact pay-out will of course vary somewhat between blocks and participants. However, the average returns can be inferred from *p,* the probability of correct response period detection (Figure 2b); *n*, the number of response periods in a block (Figure 2a); and FA, the false alarm rate during non-response periods (Figure 3a).

For example, in a typical block with response periods LONG and FREQUENT, where *N=30, p=0.9* and *FA = 2* responses/min, then the total points earned would be:

Where *n*p*3* reflects the total reward for correct responses, -1.5*(1-p)*n is the total penalty for missed responses, and -1.5 *5*FA is the total penalty for 5 minutes’ worth of false alarms. In this block, the participant would have earned (61.5/15) * 50 = £2.05, which is quite a high incentive for a five-minute task. We note that this is the condition with the highest pay-out, but the average pay-out across all four blocks (when combined across six repetitions of each block) was approximately £22. We therefore suspect that our participants were well motivated by this bonus pay-out.

To address any concerns about the reward feedback distracting the participants, we deliberately showed the reward bar only during training, and not during task performance. While playing the task during EEG recording, participants knew that the coloured dots feedback would translate into the money bar, but they were only shown their overall performance at the end of each block. We have amended the methods section to reflect this:

“A reward bar was shown at the end of each 5-minute block to indicate how many points participants have won in total (the reward bar was shown continuously onscreen during training**,**
*but not during task performance to avoid distraction*).”

– Figure 1c:" The resulting evidence integration kernel is well described by an exponential decay function, whose decay time constant in seconds is controlled by the free parameter tau ": As formulated, the caption hints that the figure analyses how well described is the kernel. As I understand, the exponential kernel is a choice that is imposed by the authors, and how well it fits the coherence level before a decision is not addressed in the paper. I would reformulate and clearly state that the authors define an exponential kernel from the past coherence levels. More details on the definition of this kernel are required, and statistics about how well it fits the data would enable stronger claims of its use throughout the text. In particular, the average coherence can take 3 values (30, 40, or 50), are the authors averaging for all coherence levels together? Should A be correlated with those values? What is the effect of the mean coherence level during response periods?

Thanks for these comments.

We should first note that although we plot integration kernels for both responses made during response periods (Figure 2d) and ‘false alarms’ (Figure 1c example, Figure 3b, Figure 3 —figure supplement 1), the exponential decay kernel is in fact only fit to ‘false alarm’ responses. The main reason for this is that the integration kernels from response periods include a shift in the mean coherence at some point between t=-5s and t=0s; this gives rise to the ‘shoulder’ in the integration kernel that can be seen in Figure 2d, and so this is not so well described by an exponential decay. We address the reviewer’s other point (about different levels of mean coherence entering into Figure 2d).

Although we had stated this point later in the original submission, we can see that it is not so clear from the very first place where we introduce the kernels in Figure 1c. We have therefore clarified it there as well:

“The resulting evidence integration kernel for false alarms is well described by an exponential decay function…”

And to the main text:

“We performed this reverse correlation for both false alarm responses (example shown in Figure 1c; these responses are well described by an exponential decay function detailed below) and correct responses.”

We also agree with the reviewer that the exponential kernel is a choice imposed by us.

However, we note that it is a theoretically-motivated choice, rather than an arbitrary one. This is because the ‘leak’ term in a leaky evidence accumulation model implies an exponential decay of previously weighted evidence across time. We have clarified this in the text:

“We note that this exponential decay model is theoretically motivated by the leaky evidence accumulation model, which implies that past evidence will leak from the accumulator with an exponential decay (Bogacz et al., 2006).”

Nevertheless, we agree that it would be worth including more statistics on how well this model fits the data, to complement the qualitative example fits that are shown in Figure 3 —figure supplement 1. To calculate the quality of the model fit, we therefore now report the *R^2^* value for the exponential decay function. We have added to the methods:

“We then fit A and τ to the empirical integration kernel for all timepoints up to and including t=0 using *fminsearch* in MATLAB, using a least squares cost function between the fitted model and data with an L2 regularisation term that penalised large values of either A or τ (l = 0.01). To calculate the quality of the model fit, we calculated R^2^ for this function: R2=1−RSSTSS with RSS being the Residual Sum of Squares after model fitting, and TSS being the Total Sum of Squares.”

And we have added this to the main text:

“Our exponential decay model provided a good fit to data at a single subject level (median R2=0.82, 95% confidence intervals for R2 = [0.42,0.93]; see Figure 3 —figure supplement 1 for example fits)…”

Our claim that the exponential decay provides a good fit to the integration kernels from false alarms is therefore now justified by the fact that it explains (on average) 82% of the variance in these integration kernels.

P7, Figure 2:– b and c: is that the mean or median (for b) and the error bar the std/95expectiles (for c) of participants' detection rate/RT?

As was already noted in the figure legend, both plots show mean +/- s.e across participants. We have added an extra subclause to the figure legend to make this even clearer:

“All plots in (b), (c) and (d) show mean +/- s.e. across participants.”

Is the RT tau?

No, RT and tau are quite different from one another. Tau relates to the fit of the exponential decay function as mentioned above, this is only calculated for false alarms, not for response periods. RT is the more standard measure of reaction time, i.e. time taken to respond during the response period after the mean coherence has changed from 0. We clarify this in the legend:

“Median reaction time (time taken to respond after start of response period) for successfully reported response periods.”

Note that the ‘median reaction time’ here refers to the median RT *within-subject* (we take the median as RTs are typically not normally distributed); as noted above, we then plot the mean +/- s.e. *across* subjects.

– d: is the integration kernel here referring to the exponential, or simply the average signal before the decision, averaged across all participants?

This is simply the average signal before the decision. We have clarified that we are only fitting the exponential decay model to ‘false alarms’ (see changes to text made above), and we also have added the following clarification:

“the sensory evidence integration properties (i.e. the ‘integration kernels’, calculated by averaging the signal prior to the decision)…”

– d again: why is the maximum coherence 0.5 here? is that only for switches from 0 to 0.5 coherence? What happens for switches from 0 to 30 coherence?

No, this plot *collapses* across all mean motion coherences (0.3, 0.4, and 0.5). We have now clarified this in the text:

“…the ‘integration kernels’, calculated by averaging the signal prior to the decision, collapsing across all levels of mean motion coherence…”

We adopt this approach because the participant is of course unaware of the level of motion coherence in the stimulus, and also the stimulus is also corrupted by noise (so in a 0.3 coherence response period, the average motion coherence can be higher than this due to noise).

In fact, it appears highly likely that the threshold shown in Figure 2d is the same used across all types of trial. This is because the same threshold is also reached in Figure 3b – responses during the *noise* periods, when the average motion coherence is zero.

– d again: the authors observe no effect on the response window duration on the response kernel, but have applied a threshold on the duration of the response on the long response period in order to make this plot. What happens if the authors include all response lengths? (also applies to the discussion p17 and 18).

If we were to include all response lengths, this does indeed give a difference in response window duration on response kernels. (If interested, we refer the reviewer to chapter 3, figure 8 of (Ruesseler, 2021), and also the preceding chapter in this doctoral dissertation). But the problem with this approach, expanded upon more extensively in simulation in this dissertation, is that it can be an entirely artefactual consequence of the change in mean signal at different latencies (due to the difference in RTs between conditions).

We therefore think that it would be misleading to include such a plot in the manuscript, and we note that reviewer #2 shares these concerns. As discussed above, the cleanest test of any difference between conditions has to be Figure 3b rather than Figure 2d, because the statistics of the stimulus are exactly matched between conditions in the noise periods rather than the response periods. This figure also shows a difference due to manipulations of response period frequency, but not response period length.

P 8:– "This provides further evidence that participants were overall more conservative in their responses in LONG conditions than SHORT" can the authors define what conservative means?

Conservative here means that for an equivalent level of sensory evidence, the participants were less likely to make a response. We can make this claim because the statistics of the noise periods are matched between LONG and SHORT conditions, but participants are less likely to false alarm in the LONG condition. We have now added to the text:

“This provides further evidence that participants were overall more conservative in their responses in LONG conditions than SHORT. (In other words, for an equivalent level of sensory evidence, the participants were less likely to make a response.)”

This is similar to the finding in response periods that participants have longer RTs in LONG than in SHORT, when matching for levels of sensory evidence (another point where we state that participants are more ‘conservative’).

– It is not surprising that the integration kernels are not affected by the length of the response period if the long responses are not considered.

Please see answer above. It would be entirely possible for this analysis to show an effect of response periods on integration kernels even without long responses being considered. This would also show up in the analysis in Figure 3b as well. This is something that we have now explored more extensively in simulation, in particular the new section on computational modelling of the Ornstein-Uhlenbeck model.

– I think the kernel analysis during baseline is necessary to show that participants are not randomly making motor movements, however, the authors could describe in a bit more detail the difference between a correct response and a false alarm: how are the integration kernels different in the baseline vs response period? (the kernels for false alarms seem a bit shorter?)

This is a good point. There are two main differences.

First, as the reviewer mentions, the kernels appear a bit shorter. This is because they are not in any way confounded by the inclusion of changes in the mean motion coherence in the analysis (see response to reviewer #2 above). Again, this means that the kernel analysis in Figure 3b is a ‘cleaner’ estimate of evidence integration in participants – which is why we fit kernels to this, use this in all our simulations, and focus on between-subject variability in these noise integration kernels.

The second difference is that right before the decision, the noise integration kernels drop towards zero, whereas the response period integration kernels stay elevated. This is because of non-decision time in the decision process. In Figure 3b, we can see that the motion coherence immediately prior to the response has less influence on the decision, because of the time needed to translate the decision into a response. In Figure 2d, by contrast, the integration kernel stays high, and this is because this time period includes the shift in average motion (i.e. the *mean* motion coherence is non-zero).

We have briefly mentioned these two main differences in the revision:

“The slight differences in integration kernels between Figure 3b and Figure 2d (shorter duration, and return to baseline close to the response) are due to the inclusion of the average motion signal in Figure 3b, rather than just the noise.”

– When do false alarms occur compared to the response periods? Are they regularly after the end of a response period?

False alarms cannot occur immediately after the end of a response period, because (as mentioned in the methods) any responses made in the 500ms after a response period were actually counted as correct responses (to allow for non-decision time in the decision process). They are also very unlikely to follow immediately after this, because the participant then gets feedback that they have missed a response, and so if anything they typically withhold responses in the period that immediately follows.

What is the pattern of the coherence that causes those false alarms? Are they always when the frame update is 1s, for example?

We are not sure what the reviewer means here.

The most obvious conclusion about the pattern of coherence that causes false alarms can be drawn from the evidence integration kernels from the noise periods, Figure 3b.

It is possible that the reviewer is asking whether responses are more likely after an intersample interval of maximum length, i.e. 1000ms. While this is an interesting idea, there are several reasons why this is very unlikely, and also difficult to analyse. Firstly, remember that the *frame* update is in fact constant throughout the entire experiment (100Hz) – the updates shown in Figure 1b are just when the underlying motion coherence changes. Secondly, the actual percept experienced by the subject is not just a function of signal plus experimenter noise (as shown in Figure 1b) – it will also have the addition of *perceptual* noise on top of this, which will presumably be constant throughout the entire experiment. Finally, even if the reviewer were correct, it would be difficult to make such a claim, due to the uncertain contribution of non-decision time in each participants’ responses (e.g. Figure 3b) – meaning that would be difficult to know exactly *which* of the inter-sample intervals preceding the response should be included in the analysis.

In short, although we can see that this is an interesting idea, we think that the most clear representation of the pattern of coherence that causes false alarms can already be observed from Figure 3b (and associated Figures).

Figure 3:a) the authors mention significance but do not provide p values.

We previously provided an F-statistic and associated p-value in the main text:

F(1,23)=58.67, p=8.98*10^-8^. We have now copied this into the figure legend.

Why is it consistent with having a more cautious response threshold? I'd favor the reverse interpretation that short response periods induce more confusion between signal and noise.

Thanks for suggesting this alternative interpretation. We have added this idea to figure legend:

“This is consistent with having a more cautious response threshold (also evidenced by longer reaction times during response periods, main Figure 3c), although could also be interpreted as shorter response periods inducing more confusion between signal and noise.”

b) is that the averaged signal before false alarms?

Yes.

Do they always occur when the mean average coherence reaches 0.5?

No, this is the average across many false alarms (which will have differing values).

Why is the maximum value 0.5?

This is just a consequence of the behaviour of the subjects – it shows on average the strength of signal that was needed to trigger a response (and is entirely consistent with what is found in response periods, Figure 2d). As shown in our new simulations of the Ornstein-Uhlenbeck process, there is no *a priori* reason why the maximum value has to be 0.5, as it will depend upon the internal parameters of the decision process (and indeed, it varies across participants, as can be seen in Figure 3 —figure supplement 1).

c) explain what the points show and how these are computed.

We have now added to the figure legend:

“The data points show the time constant, τ , for each participant after fitting a model of exponential decay to the integration kernel. The equation for this kernel is in the main text, and details of kernel fitting are provided in Methods.”

P.11(ii) To me, it is not clear that the behavioral evidence from 'false alarms' indicate that participants are still integrating sensory evidence – a stricter definition and existence of the kernel (how well it fits for every response) could help support this statement

While we agree with the spirit of this comment, it would be very difficult to measure exactly how the kernel fits for every single response, without a clear articulation of what the control model might be in this comparison. We have, however, considered elsewhere what impact a strategy like ‘peak detection’ might have on the resulting evidence integration kernels, and how this would compare with what we observe in the data. If the reviewer is interested, this can be examined by looking at chapter 3, figure 4 of (Ruesseler, 2021), which shows that such a strategy elicits integration kernels that look quite different from those we observe in the data.

– Is 'net motion' coherence here?

Yes – thanks, we have changed this to align it with other mentions of ‘coherence’ elsewhere in the text.

– On the same page – the regressor 'absoluted sensory evidence' has not been named like this before.

Thanks – we have renamed this to |evidence|, to make it clearer that we are referring to the same regressor as introduced earlier on the same page.

P. 12:– Why is it consistent with the behavioral finding of lower detection rate and slower RT in rare conditions?

We have now expanded on this conclusion, and made clear how it relates to the findings in our computational modelling section above:

“In addition, when making a correct response, preparatory motor activity over central electrodes reached a larger decision threshold for RARE vs. FREQUENT response periods (Figure 7b; p=0.041, cluster-based permutation test). We found similar effects in β-band desynchronisation prior, averaged over the same electrodes; β desynchronisation was greater in RARE than FREQUENT response periods. As discussed in the computational modelling section above, this is consistent with the changes in integration kernels between these conditions as it may reflect a change in decision threshold (Figure 2d, 3c/d). It is also consistent with the lower detection rates and slower reaction times when response periods are RARE (Figure 2 b/c), which also imply a higher response threshold.”

– I understand the analysis is done regardless of the response. What about missed response periods/responses during the baseline?

As discussed in the methods, we include regressors for responses during the baseline, to ensure that we separately model out the EEG response to making a buttonpress during this period. Missed response periods are less relevant, because there is no additional event to model out of the EEG data (as there is no response being made).

Figure 4:– The caption is incomplete, what does the right-hand side show? Is it one weight for one regressor across time for one example electrode?

Thanks – we have now clarified the caption:

“This leads to a set of temporal response functions for each regressor at each sensor, shown on the right-hand side of the figure. The timecourse for each regressor shows the average regression weights at the three sensors highlighted with triangles on the scalp topography.”

– What is the goodness of fit of the regression?

In this case, a measure of overall goodness-of-fit of the regression (such as R^2^) is unlikely to be a useful measure, and will be very low – this is simply because the residuals (unexplained variance) in our model will remain very large at a single subject level (simply due to the low SNR of EEG data, and the contribution of many other factors to neural responses). For this reason, the goodness-of-fit of lower-level GLMs is rarely considered in either convolutional (or conventional ERP) studies of EEG data, or standard event-related analysis of fMRI data. Instead, the conventional way to assess reliability of the regression model fit is to assess the consistency of the β coefficients across participants, which we do by plotting the error bars (across participants) e.g. in Figures 7, Figure 7 —figure supplement 1, Figure 8.

P13 – Figure 5:– What are the lines showing? What is the black thick line on top?

Apologies, we should have been clearer in this figure legend. The thick black line on top is in fact a large number of * symbols super-imposed. We have now added:

“Lines and error bars show mean +/- s.e.m. across participants. * (solid black line at top of figure) denotes significant difference between FREQUENT and RARE (p<0.05, cluster corrected for multiple comparisons across time).”

– P.14 To test whether these signals are decision-related, the authors could also compare the results of the regression when false alarms are made and when the response period is missed.

We found qualitatively similar results for false alarms, but the results were somewhat noisier (simply because there are more responses made during response periods than during false alarms). We now mention this in the text.

– Figure 6: has the effect of previous exposure been controlled for the vertical motion? Could the authors discuss whether this would have an impact on the CPP.

During training for the combined horizontal/vertical motion task, participants were exposed to both horizontal and vertical motion in the training stimuli. We therefore consider this to be an unlikely explanation for the differences shown in Figure 6 (now figure 8).

– P 15: could the authors give more details on the behavioral correlate and how is it computed.

Certainly. We have added a sentence to clarify where this behavioural correlate is coming from:

“We therefore performed a behavioural-neural correlation between participants’ integration time constants τ and their TRFs to sensory noise fluctuations. (Note that the integration time constants were fit using the equation described above, fit (using the approach described in methods) separately to the empirical integration kernels from each of the four conditions).”

– "this negative-going component was larger in amplitude (i.e. more negative) in participants who would integrate sensory evidence over longer durations (i.e. had a higher value of τ).": could the authors elaborate on how to interpret this finding with regard to existing literature?

We have added the following comment to the manuscript:

“We suggest that this may be consistent with variation in the encoding strength of previously studied correlates of continuous decision evidence. For example, Wyart et al. found a positive centroparietal potential 500ms after decision information that positively encoded the current sample, but negatively encoded adjacent samples (Wyart et al., 2012); our finding extends this work to explore variation in the response across participants.”

– Figure 7 P 16 spearman correlation: can the authors describe the top plot? what are the lines showing? if that's the mean it should be mentioned and the standard deviation should be shown.

Perhaps the reviewer missed it, but this was already described in the figure legend. The lines show Spearman’s rho (as also mentioned on the y-axis of the plot), not the mean.

“Top panel shows Spearman’s rank correlation between the time-varying EEG β for absolute sensory evidence and individual subjects’ τ parameter, separately for each of the four conditions.”

Can the authors describe the bottom plots? why are the authors using log(tau) for the x-axis?

We now clarify this in the figure legend.

“We plot the average EEG effect size against log(τ) to allow for a straight-line fit (lines show mean +/- 95% confidence intervals of a first-order polynomial fit between these two variables); we used Spearman’s rho to calculate the relationship, as it does not assume linearity.”

Discussion:- p 17: 'The behavior is well described by a leaky evidence accumulator' the exponential kernel is a reverse correlation that the authors have done themselves, but they have not shown causality nor statistical analysis on what those kernels mean in relationship with the behavior.

In response to the reviewer’s earlier comment, we have now added statistics to justify our claim of the goodness of fit of the exponential kernel. We have reworded the sentence to remove the claim of it being an accumulator, and instead refer just to the kernel:

“We found that participants' behaviour was well described by an exponentially decaying integration kernel, and that participants adapted the properties of this process to the overall statistics of the sensory environment (Ossmy et al., 2013) across four different experimental conditions.”

Methods:– In the regressor table, what is 'prediction error' regressor?

Thanks for spotting this – this had slipped through from an earlier naming convention that we used in the lab for these regressors, before we decided on a more precise set of names for them. We have now renamed these in the regressor table so that they are consistent with the rest of the paper.

We thank for reviewer again for the detailed feedback on our paper, and we hope that we have successfully addressed their comments and concerns with our revision.